# Distinct roles of monkey OFC-subcortical pathways in adaptive behavior

Kei Oyama [1,2], Kei Majima [2,3], Yuji Nagai [1], Yukiko Hori [1], Toshiyuki Hirabayashi [1], Mark A. G. Eldridge [4], Koki Mimura [1,5], Naohisa Miyakawa[1], Atsushi Fujimoto [1], Yuki Hori[1], Haruhiko Iwaoki[1], Ken-ichi Inoue [6], Richard C. Saunders[4], Masahiko Takada [6], Noriaki Yahata[3], Makoto Higuchi [1], Barry J. Richmond [4] & Takafumi Minamimoto [1] ✉

Primates must adapt to changing environments by optimizing their behavior to make beneficial choices. At the core of adaptive behavior is the orbitofrontal cortex (OFC) of the brain, which updates choice value through direct experience or knowledge-based inference. Here, we identify distinct neural circuitry underlying these two separate abilities. We designed two behavioral tasks in which two male macaque monkeys updated the values of certain items, either by directly experiencing changes in stimulus-reward associations, or by inferring the value of unexperienced items based on the task's rules. Chemogenetic silencing of bilateral OFC combined with mathematical model-fitting analysis revealed that monkey OFC is involved in updating item value based on both experience and inference. In vivo imaging of chemogenetic receptors by positron emission tomography allowed us to map projections from the OFC to the rostromedial caudate nucleus (rmCD) and the medial part of the mediodorsal thalamus (MDm). Chemogenetic silencing of the OFC-rmCD pathway impaired experience-based value updating, while silencing the OFC-MDm pathway impaired inference-based value updating. Our results thus demonstrate dissociable contributions of distinct OFC projections to different behavioral strategies, and provide new insights into the neural basis of value-based adaptive decision-making in primates.

To survive in a constantly changing world, animals naturally adapt quickly to new environments and adjust their behavior to maximize the benefits. This involves making decisions that will lead to maximum subjective benefit, based on the changing relationships between specific events and outcomes, and thus requires knowing the current worth of each option when making a choice. Typically, an item's worth is learned through direct experience, a process often explained through the concept of classical reinforcement learning[1]. However, animals with highly developed brains, particularly primates, have also evolved the ability to infer the value of unexperienced events/items from their knowledge of the world. This ability is described by the rule or theory of shifting relationships. For example, if a monkey is eating a banana and notices that it is ripe, it may be able to infer that other nearby bananas are also ripe, based on its knowledge of how fruits ripen. Optimal decision-making relies on a balance between experience- and inference-based behavioral strategies.

[1]Advanced Neuroimaging Center, National Institutes for Quantum Science and Technology, Chiba, Japan. [2]PRESTO, Japan Science and Technology Agency, Kawaguchi, Japan. [3]Institute for Quantum Life Science, National Institutes for Quantum Science and Technology, Chiba, Japan. [4]Laboratory of Neuropsychology, National Institute of Mental Health, National Institutes of Health, Bethesda, MD, USA. [5]Research Center for Medical and Health Data Science, The Institute of Statistical Mathematics, Tachikawa, Japan. [6]Systems Neuroscience Section, Center for the Evolutionary Origins of Human Behavior, Kyoto University, Inuyama, Japan. ✉e-mail: minamimoto.takafumi@qst.go.jp

The primate orbitofrontal cortex (OFC) is thought to contribute to such adaptive behavior by leveraging both direct experience and knowledge of the current context. The OFC has long been thought to play an essential role in encoding the subjective values of alternative events/items that guide subsequent decision-making, and in learning/updating these values by integrating past experiences as consequences of our choices[2-6]. At the same time, the OFC has also been shown to be necessary for inferring value based on mental simulation of outcomes, even in the absence of direct experience, by generalizing knowledge of the current situation or environment[7-9]. A recent report suggests that the OFC regulates the balance between these two valuation strategies, rather than simply initiating one or the other[10]. Thus, there is ongoing debate about the core function of the OFC in adaptive behavior.

The complexity of OFC function might arise from its interactions with other brain regions through direct anatomical connections[11]. For example, subcortical structures, such as the rostromedial part of the caudate nucleus (rmCD) and the medial part of the mediodorsal thalamus (MDm), receive direct projections from the OFC[12-14]. Lesions to these areas have been shown to produce deficits that are similar, but not identical, to those produced by OFC lesions[15-20] with a tendency that rmCD and MDm are particularly involved in value-updating based on experience and inference, respectively, suggesting that they have overlapping yet distinct roles in adaptive behavior. These findings have raised the possibility that two pathways originating from OFC, namely the OFC-rmCD and OFC-MDm pathways, are needed for different value-updating strategies. To investigate this possibility, it is essential to independently manipulate the prefronto-subcortical circuits, which is technically challenging, especially in behaving nonhuman primates.

To investigate the causal roles of the OFC and its originating pathways in these two types of valuation, we use a chemogenetic tool called designer receptors exclusively activated by designer drugs (DREADDs)[21]. This tool allows neurons to be silenced by activating an inhibitory DREADD (hM4Di) following systemic administration of a DREADD agonist. Additionally, local agonist infusion that activates hM4Di expressed at axon terminals can suppress synaptic transmission[22,23]. By combining these techniques with positron emission tomography (PET) as an in vivo imaging tool for hM4Di-positive projection sites, we have previously developed an *imaging-guided chemogenetic synaptic-silencing* method that is dramatically more efficient and accurate, especially when applied to nonhuman primates[24]. Leveraging this technique and a model-fitting analysis in a reinforcement learning framework, the present study addresses the contributions of these two OFC-subcortical pathways to different value-updating strategies. Our results suggest that experience- and inference-based strategies for updating stimulus-reward associations rely on the OFC-rmCD and OFC-MDm pathways, respectively.

## Results

### Experience- and inference-based behavioral adaptation in multi-reward value-updating tasks

To address the question of how the OFC and its projections to subcortical structures contribute to behavioral adaptation through experience- and inference-based valuation, we devised two behavioral tasks for macaque monkeys: NOVEL and FAMILIAR tasks, respectively (Fig. 1a). In both tasks, the monkeys were required to choose either of two presented visual stimuli (out of a set of five abstract images), each of which was associated with 1, 2, 3, 4, or 5 drops of juice. The order of associations was reversed within a session (Fig. 1a). To maximize their reward, monkeys had to learn the values of the visual stimuli and then update them following subsequent changes in stimulus-reward associations. In the NOVEL task, which was aimed at assessing experience-based updating, a new set of stimuli was introduced in each session, thus requiring the monkeys to learn new stimulus-reward associations, as well as the association reversal that was imposed mid-session (90

trials after the beginning of each session). After several months of training, two monkeys (Mk#1 and Mk#2) were able to learn new stimulus-reward associations within 80–90 trials and adapted to their reversal within 30–50 trials (Fig. 1b, right). In the FAMILIAR task, which was aimed at assessing inference-based updating, a fixed set of five visual stimuli was used throughout the experiments, and the reversals were imposed several times after performance reached a predetermined criterion (see Methods). Following several months of training on this task, the monkeys were able to adapt to the reversal within 3–5 trials (Fig. 1c, right). They even showed optimal choice for "unexperienced" stimulus-reward associations after experiencing the other associations following the reversal (for details, see the last paragraph of the Results section), indicating that they solved this task based on inference, that is, their prior knowledge of the limited patterns of stimulus-reward associations.

To examine whether the monkeys solved these tasks based on experience- or inference-based strategies, we conducted simulations using two types of reinforcement learning models (see Methods for details); one assuming that the values were updated through direct experience based on standard model-free reinforcement learning that was driven by reward prediction errors ("EXP" model), and one assuming that the monkeys had a priori knowledge of the two possible stimulus-reward association patterns (1,2,3,4,5 or 5,4,3,2,1 drops) through daily training, and that prediction errors drove the transition between these two already learned value sets, allowing them to infer the values of any unexperienced stimuli ("INF" model). As expected, this analysis revealed that behavior during the NOVEL task was better explained by the EXP model (Fig. 1d), while that during the FAMILIAR task was better explained by the INF model (Fig. 1e). These results suggest that monkeys "solved" the two tasks with different strategies − experience-based for the NOVEL task and inference-based for the FAMILIAR task.

### OFC silencing impairs both experience- and inference-based strategies

Next, we chemogenetically inactivated the OFC to determine whether it contributes to experience- and/or inference-based valuation strategies. First, we bilaterally introduced the inhibitory DREADD hM4Di into the OFC (Brodmann's area 11/13, hereafter referred to as "OFC[11/13]") of each monkey via injections of an adeno-associated virus (AAV) vector (AAV2-CMV-hM4Di and AAV2.1-CaMKII-hM4Di-IRES-AcGFP for Mk#1 and Mk#2, respectively) (Fig. 2a). Several weeks after the injections, we noninvasively visualized hM4Di expression using PET imaging with DREADD-PET tracers. In both monkeys, we consistently observed increased PET signal in the bilateral OFC (Fig. 2a; Supplementary Fig. 1a), which was confirmed by post-mortem immunohistochemistry to reflect hM4Di expression. We then examined the effect that silencing OFC[11/13] had on behavior; task performances were compared after systemically administering either a vehicle control or the DREADD agonist deschloroclozapine (DCZ) (Fig. 2b). While silencing OFC[11/13] did not alter performance on the NOVEL task before reversal of the stimulus-reward contingencies (acquisition phase), it did impair performance following the reversal, which was particularly true in the early phase (100 post-reversal trials, PoE in Fig. 2c, e). Similarly, OFC[11/13] silencing also impaired performance on the FAMILIAR task after reversals (Fig. 2d, g). To quantify the silencing effect in both tasks, we fitted two behavioral models to the behavioral data following DCZ and vehicle administration. Model-fitting analysis revealed that the impaired performance after OFC[11/13] silencing could be attributed to a decrease in the learning rate (α) after the reversals (Fig. 2f, h, see Methods for details), but not to the change in the extent of exploration (the inverse temperature, β; Supplementary Fig. 2a, d), suggesting that silencing OFC[11/13] led to deficits in value-updating when using either strategy.

We conducted four additional experiments to confirm that the observed behavioral changes were due to the loss of normal OFC[11/13] function that is involved in adapting to the shift in the task context, and

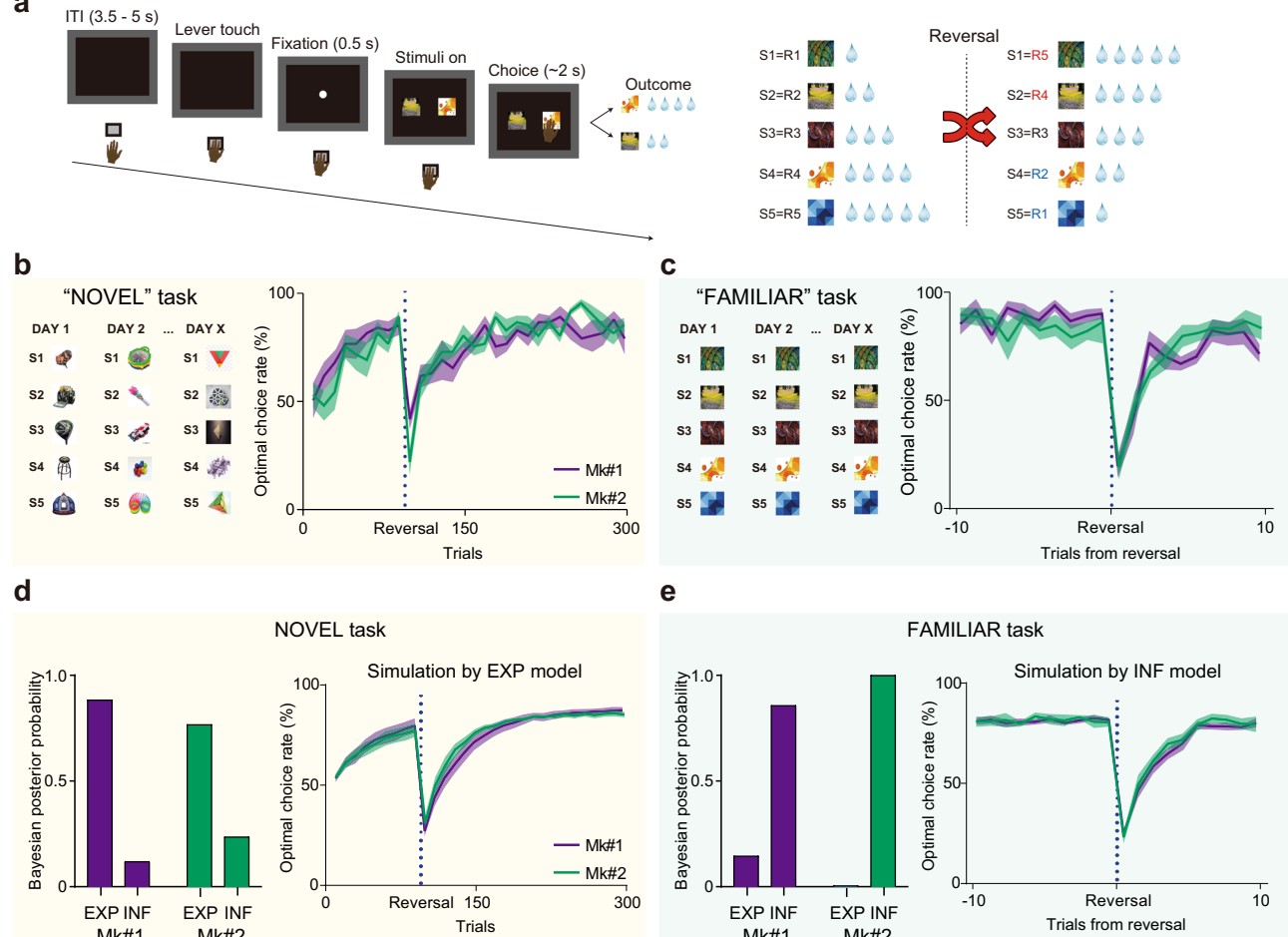

**Fig. 1 | Experience- and inference-based value updating in multi-reward reversal learning tasks. a** Sequence of a trial (left) and the reversal rule for stimulus-reward associations (right). S1-S5 represent the identity of each stimulus and R1-R5 represent the amount of reward (1 to 5 drops of juice) associated with each stimulus. Examples of stimulus sets and baseline performance for the "NOVEL" (**b**) and "FAMILIAR" (**c**) tasks. Averaged performance for each monkey is presented ($N = 8$ and 7 sessions for the NOVEL and FAMILIAR tasks, respectively). Data for the

FAMILIAR task were truncated to show those around the reversals and were averaged across reversals. Bayesian posterior probability calculated for "EXP" and "INF" models given the behavioral data in each task (left) and the behavioral simulation by the model with higher posterior probability for each monkey (right); EXP model for the NOVEL task (**d**) and INF model for FAMILIAR task (**e**), respectively. Solid lines and shaded area represent the mean and s.e.m, respectively.

which requires updating the values of the external stimuli. First, to confirm that healthy OFC[11/13] function is required at the time of the reversal, but not when learning the stimulus-reward associations or during any other process in the acquisition phase, we administered DCZ after the acquisition phase (just before reversal, see Methods for details). This manipulation produced deficits similar to those observed when DCZ was administered before the beginning of each session (Supplementary Fig. 3a, b), suggesting that the effects of OFC[11/13] silencing were limited to the reversal of stimulus-reward associations. Second, we confirmed that DCZ alone did not significantly affect behavioral performance before the introduction of hM4Di (Mk#2, Supplementary Fig. 3c, d), indicating that the effects observed after DCZ administration were due to DREADD activation. Third, we confirmed that OFC[11/13] silencing did not significantly impact simple reversal learning (Supplementary Fig. 4), consistent with a previous lesion study[25]. Fourth, to test whether the OFC[11/13] is also essential for situations in which knowledge-based value-updating is required and the cognitive load is high, but the item values are binary (i.e., reward or no reward), we examined the effects of OFC[11/13] silencing using an analog of the Wisconsin Card Sorting Test[26] (Mk#2, Supplementary Fig. 5b). In this case, we found that silencing OFC[11/13] had no effect on task performance (Supplementary Fig. 5c), suggesting that it is

required specifically when updating external items with complex, non-binary values. Importantly, chemogenetic silencing of OFC[11/13] remained effective at the end of all experiments, including these control experiments and the pathway-selective manipulations (see below) (Supplementary Fig. 5a), as demonstrated by the significant effects of DCZ administration on performance during the final devaluation task (Supplementary Fig. 5d, e), which is one of the most common tasks requiring normal OFC function[7,27]. Taken together, these results suggest that OFC[11/13] is essential for adapting behavioral responses that are specifically contingent upon being able to update multiple values of multiple external stimuli.

**The OFC[11/13]-rmCD and OFC[11/13]-MDm pathways are necessary for experience- and inference-based value-updating, respectively**

Having demonstrated that OFC[11/13] is essential for both experience- and inference-based behavioral adaptation, the next question is whether these different strategies are governed by separate neural pathways. To answer this, we conducted a chemogenetic pathway-selective manipulation wherein information flow can be temporarily inactivated by local agonist infusion into axonal terminals expressing hM4Di. It is generally challenging to precisely localize and target axonal projection sites in vivo in monkeys, which have relatively large and complexly shaped

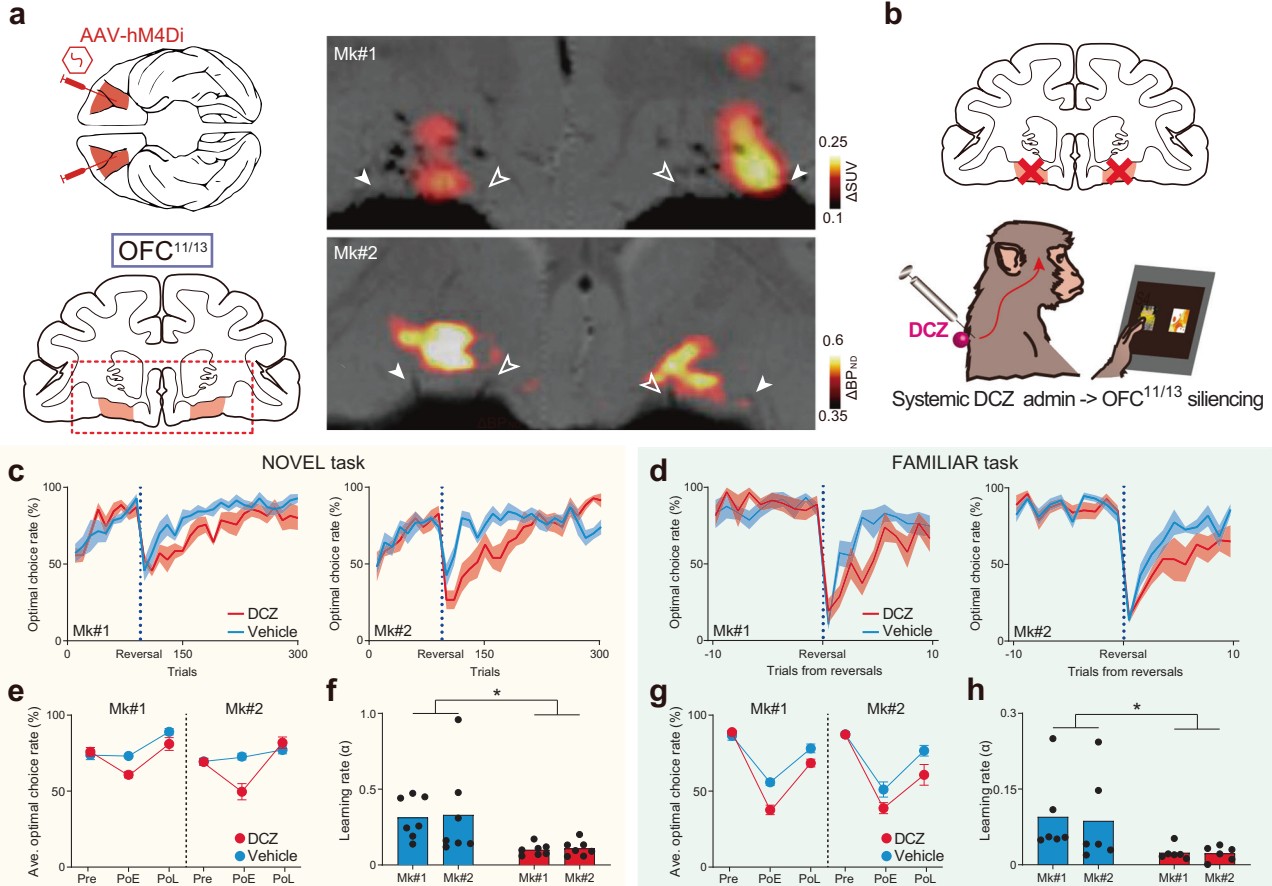

**Fig. 2 | Chemogenetic silencing of bilateral OFC[11/13] impaired experience- and inference-based value updating. a** Injection area (bilateral OFC, Brodmann's area 11/13) and PET images showing hM4Di expression in each monkey. Filled and open arrowheads represent the lateral and medial orbital sulcus, respectively. **b** Shema of OFC[11/13] silencing per se. DCZ (100 μg/kg) was systemically injected intramuscularly. **c**, **d** Behavioral effects of chemogenetic OFC[11/13] silencing on NOVEL and FAMILIAR (**d**) task performance for each monkey. Data for vehicle injections (cyan) and DCZ injections (red) are shown. The optimal choice rate following DCZ (red) and vehicle injections (cyan) was averaged across two monkeys. Solid lines and shaded area represent the mean and s.e.m, respectively. Averaged optimal choice rate for each phase of reversal (Pre, Pre-reversal; PoE, PoL, early, and late phase of the post-reversal trials; see Methods) for each monkey in the NOVEL (**e**) and FAMILIAR (**g**) tasks. In the NOVEL task, a three-way ANOVA (subject × phase × treatment) revealed a significant main effect of treatment ($F_{(1,72)} = 12.3$, $p = 7.9 \times 10^{-4}$) and a significant interaction between phase and treatment ($F_{(2,72)} = 11.0$, $p = 6.8 \times 10^{-5}$). Subsequent two-way ANOVAs (subject × treatment) for each phase revealed significant

differences for treatment during the PoE ($F_{(1,24)} = 30.0$, $p = 1.2 \times 10^{-5}$), but not during the Pre ($F_{(1,24)} = 0.11$, $p = 0.74$) or the PoL ($F_{(1,24)} = 0.25$, $p = 0.62$). Similarly, in the FAMILIAR task, a three-way ANOVA revealed a significant main effect of treatment ($F_{(1,60)} = 20.8$, $p = 2.6 \times 10^{-5}$) and significant interaction between phase and treatment ($F_{(2,60)} = 6.6$, $p = 2.5 \times 10^{-3}$). The two-way ANOVAs revealed significant differences for treatment during the PoE ($F_{(1,20)} = 17.8$, $p = 4.2 \times 10^{-4}$) and the PoL ($F_{(1,20)} = 8.9$, $p = 7.2 \times 10^{-3}$), but not during the Pre ($F_{(1,20)} = 0.32$, $p = 0.58$). Error bars: s.e.m. Estimated learning rates in the NOVEL task with the EXP model (**f**, two-way ANOVA, subject × treatment, main effect of treatment, $F_{(1,24)} = 11.3$, $p = 2.6 \times 10^{-3}$; subject, $F_{(1,24)} = 0.04$, $p = 0.84$; interaction, $F_{(1,24)} = 0.004$, $p = 0.95$)) and in the FAMILIAR task with the INF model (**h**, treatment, $F_{(1,20)} = 7.5$, $p = 1.3 \times 10^{-2}$; subject, $F_{(1,20)} = 0.04$, $p = 0.85$; interaction, $F_{(1,20)} = 0.02$, $p = 0.90$). Asterisks: $p < 0.05$ for significant main effect of treatment. Note that the learning rates for the NOVEL and FAMILIAR tasks were calculated using different models. Thus, they are not directly comparable. Data were obtained from $N = 7$ and 6 sessions for each treatment in each monkey for the NOVEL and FAMILIAR tasks, respectively.

brains. We overcame this obstacle by using an imaging-guided chemogenetic technique[24] in which PET allows localization of the projection sites for hM4Di-positive OFC[11/13] neurons. Aside from OFC[11/13], subtraction PET images (post-AAV injection minus pre-AAV injection) showed increased PET signals in the striatum and the thalamus, specifically in the rmCD, MDm, and the medial part of the putamen (Fig. 3a, b; Supplementary Fig. 1b–d). These regions colocalized with GFP-positive axon terminals under immunohistological examination (Supplementary Fig. 1b–d). In contrast, the hM4Di signal was not clearly observed via PET or histology in other brain regions that are known to also receive projections from OFC[11/13], such as the amygdala, and was not comparable to what we observed in these three regions (Supplementary Fig. 1e). Thus, our next experiments focused on the projections from OFC[11/13] to the three DREADD-positive terminal regions.

To reversibly block neuronal transmission from OFC[11/13], we infused DCZ into one of the three terminal regions under the guidance

of MR, PET, and CT imaging (Fig. 3a, b). Compared with the control vehicle infusion, local DCZ infusion into the rmCD impaired performance on the NOVEL task just after reversals, similar to the systemic DCZ injections (Fig. 3c, e, left; effect size of treatment, 0.50 vs. 0.48 for systemic injections and local infusions into the rmCD). However, this was not the case for the FAMILIAR task (Fig. 3c, e, right). Conversely, DCZ infusion into the MDm did not impair performance on the NOVEL task (Fig. 3d, f, left), but did impair performance on the FAMILIAR task (Fig. 3d, f, right; effect size of treatment, 0.46 vs. 0.29 for systemic injections and local infusions into the MDm). The effect of local DCZ infusion appeared to be consistent throughout the session, with no systematic differences in task performance between the 1st and 2nd halves (Supplementary Fig. 6). Although we also injected DCZ into the medial part of the putamen, where we found weak PET and histological signals (Supplementary Fig. 1d), this did not affect performance on the NOVEL task (Supplementary Fig. 7a, b). In addition, control DCZ

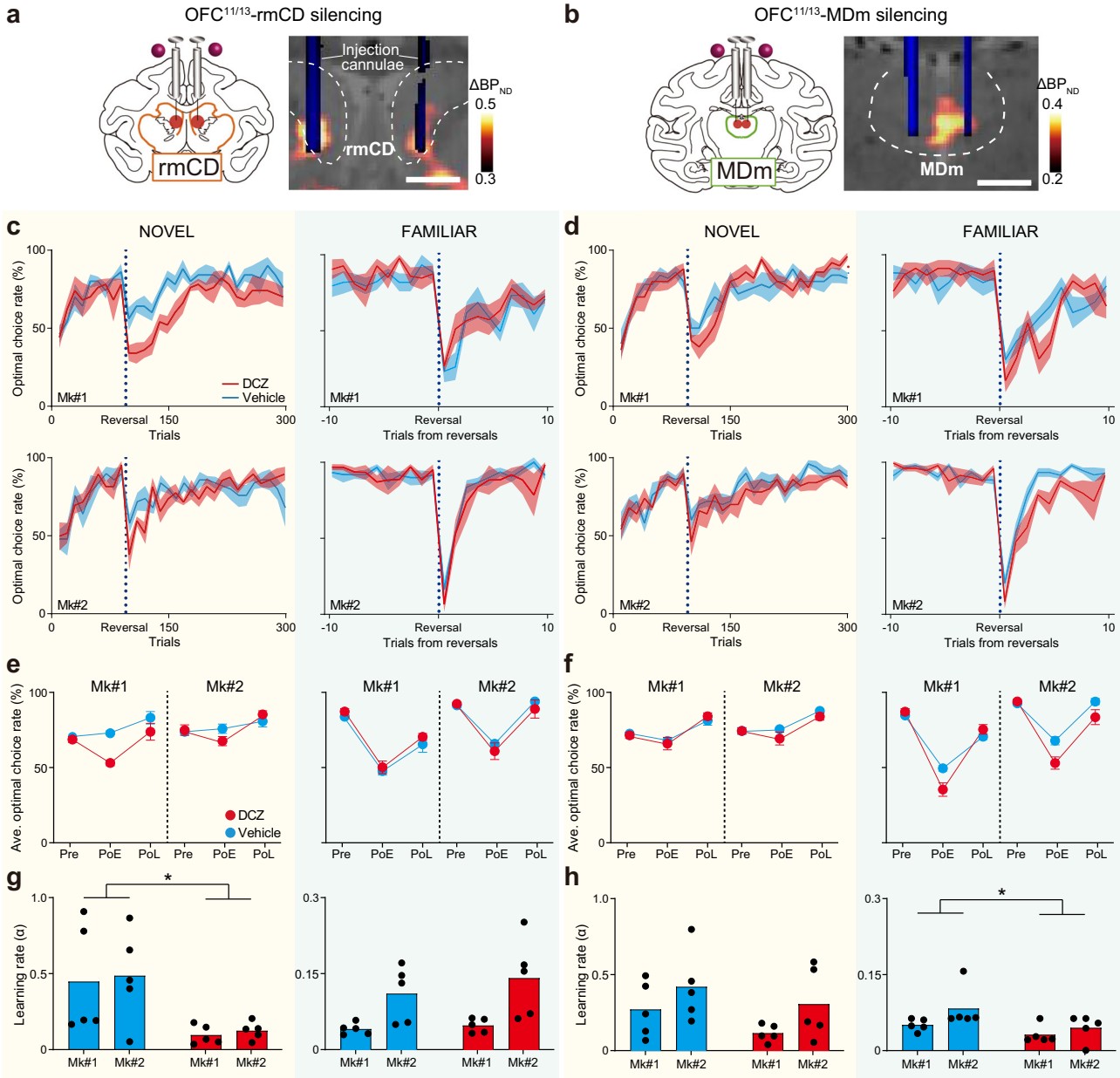

infusions into the hM4Di-negative brain region (anterior thalamus; the structure between the rmCD and the MDm) had no significant impact on behavior in the NOVEL task (Supplementary Fig. 7c, d), thus supporting the claim that our data described above are attributable to the chemogenetic silencing of hM4Di-positive OFC[11/13]-derived axon terminals. We therefore focused further tests on the OFC[11/13] to rmCD and OFC[11/13] to MDm pathways. The order of the experiments, including systemic injections and local infusions into each area, for each monkey is summarized in Supplementary Fig. 8.

Model fitting analysis using two reinforcement learning models revealed that silencing the OFC[11/13]-rmCD pathway significantly reduced the learning rate during the NOVEL task (EXP model; Fig. 3g, left), but not during the FAMILIAR task (INF model; Fig. 3g, right). Conversely, silencing the OFC[11/13]-MDm pathway had no impact on the learning rate during the NOVEL task (Exp model; Fig. 3h, left), but significantly reduced it during the FAMILIAR task (INF model; Fig. 3h, right). Similar to OFC[11/13] silencing, the inverse temperature was not affected by silencing either pathway (Supplementary Fig. 2b, c, e, f).

Damage to the OFC in humans has been associated with increased impulsivity[28]. In monkeys, OFC inactivation or lesioning has resulted in

faster reaction times on experimental tasks[29], which is generally interpreted as a sign of impulsivity or lack of control. To assess whether silencing OFC[11/13] and its projections affected impulsivity in our task context, and if so, whether this behavioral change is related to the observed impairment in performance, we examined reaction times after each type of silencing (Supplementary Fig. 9). OFC[11/13] silencing resulted in shorter reaction times for both monkeys during all task phases, but we did not observe any direct relationship between reaction time and performance (Supplementary Fig. 9a). In contrast, silencing the OFC[11/13]-rmCD and OFC[11/13]-MDm pathways induced complex and contradicting results; silencing the OFC[11/13]-rmCD pathway increased reaction time (Supplementary Fig. 9b), whereas silencing the OFC[11/13]-MDm pathway did not influence reaction time (Supplementary Fig. 9c). Although these effects on reaction time differed, we can conclude that the difficulty in updating item values was unrelated to increased impulsivity.

Taken together, the outcomes after selectively silencing the two OFC[11/13] projections indicated that both are involved in updating stimulus values; the OFC[11/13]-rmCD pathway is needed when updating based on direct experience of stimulus-reward associations, whereas

**Fig. 3 | OFC$^{11/13}$-rmCD and OFC$^{11/13}$-MDm pathways are necessary for experience- and inference-based value-updating, respectively.** Chemogenetic silencing of the OFC$^{11/13}$-rmCD (**a**) and OFC$^{11/13}$-MDm (**b**) pathways by local DCZ infusion into either bilateral rmCD or MDm, specifically at hM4Di-positive OFC terminal sites. A CT image showing the infusion cannulae (blue) overlaying a structural MR image (gray), and a PET image showing a high [11 C]DCZ binding region (hM4Di expression, hot color) obtained from Mk#2. The dashed lines represent the borders of the caudate nucleus and mediodorsal thalamus, respectively. **c,d**, Optimal choice rate in the NOVEL (left) and FAMILIAR (right) tasks after silencing the OFC$^{11/13}$-rmCD (**c**) and OFC$^{11/13}$-MDm (**d**) pathways. Solid lines and shaded area represent the mean and s.e.m, respectively. Averaged optimal choice rate for each phase in the NOVEL (left) and FAMILIAR (right) tasks after silencing the OFC$^{11/13}$-rmCD (**e**) and OFC$^{11/13}$-MDm (**f**) pathways. For silencing the OFC$^{11/13}$-rmCD pathways, a three-way ANOVA (subject × phase × treatment) revealed a significant main effect of treatment ($F_{(1,48)} = 9.6$, $p = 3.2 \times 10^{-3}$) and a significant interaction between phase and treatment ($F_{(2,48)} = 5.5$, $p = 7.1 \times 10^{-3}$) in the NOVEL task, but not in the FAMILIAR task (treatment, $F_{(1,48)} = 0.02$, $p = 0.88$; interaction, $F_{(2,48)} = 0.26$, $p = 0.77$). Subsequent two-way ANOVAs (subject × treatment) for each phase revealed significant differences for treatment during the PoE ($F_{(1,16)} = 32.2$, $p = 3.4 \times 10^{-5}$), but not during the Pre ($F_{(1,16)} = 0.03$, $p = 0.86$) or the PoL ($F_{(1,16)} = 0.33$, $p = 0.57$) of the NOVEL task. Note that there was a significant interaction during the PoE ($F_{(1,16)} = 5.5$, $p = 3.2 \times 10^{-2}$), with a significant difference (Mk#1, $t_{(8)} = 8.5$, $p = 3.7 \times 10^{-5}$) and a tendency (Mk#2,

$t_{(8)} = 1.9$, $p = 0.09$), as determined by individual Welch's t-tests. After silencing the OFC$^{11/13}$-MDm pathways, a three-way ANOVA (subject × phase × treatment) revealed a significant main effect of treatment ($F_{(1,48)} = 9.4$, $p = 3.6 \times 10^{-3}$) and a significant interaction between phase and treatment ($F_{(2,48)} = 8.5$, $p = 7.2 \times 10^{-4}$) in the FAMIL-IAR task, but not in the NOVEL task (treatment, $F_{(1,48)} = 1.2$, $p = 0.27$; interaction, $F_{(2,48)} = 0.60$, $p = 0.55$). Subsequent two-way ANOVAs (subject × treatment) for each phase revealed significant differences for treatment during the PoE ($F_{(1,16)} = 18.1$, $p = 6.1 \times 10^{-4}$), but not during the Pre ($F_{(1,16)} = 1.1$, $p = 0.31$) or the PoL ($F_{(1,16)} = 0.73$, $p = 0.41$) of the FAMILIAR task. Error bars: s.e.m. Estimated learning rates after silencing the OFC-rmCD pathway during the NOVEL (**g**, left, two-way ANOVA, treatment, $F_{(1,16)} = 11.0$, $p = 4.4 \times 10^{-3}$; subject, $F_{(1,16)} = 0.10$, $p = 0.76$; interaction, $F_{(1,16)} = 0.0022$, $p = 0.96$) and FAMILIAR (**g**, right, treatment, $F_{(1,16)} = 0.74$, $p = 0.40$; subject, $F_{(1,16)} = 14.1$, $p = 1.8 \times 10^{-3}$; interaction, $F_{(1,16)} = 0.30$, $p = 0.59$) tasks, and the OFC-MDm pathway during the NOVEL (**h**, left, treatment, $F_{(1,16)} = 2.5$, $p = 0.14$; subject, $F_{(1,16)} = 3.9$, $p = 0.07$; interaction, $F_{(1,16)} = 0.059$, $p = 0.81$) and FAMILIAR (**h**, right, treatment, $F_{(1,16)} = 5.7$, $p = 2.9 \times 10^{-2}$; subject, $F_{(1,16)} = 3.7$, $p = 0.07$; interaction, $F_{(1,16)} = 0.61$, $p = 0.45$) tasks. Additional analysis using three-way ANOVA (subject × treatment × injection area) during the Po1 phase revealed significant interactions between treatment and injection area in both tasks (NOVEL, $F_{(1,32)} = 6.1$, $p = 1.9 \times 10^{-2}$; FAMILIAR, $F_{(1,32)} = 6.6$, $p = 1.5 \times 10^{-2}$), indicating that the effects of DCZ infusion into different areas were significantly different. Data were obtained from $N = 5$ sessions for each treatment, each task, and each monkey. Scale bars: 5 mm.

the OFC$^{11/13}$-MDm pathway is needed when updating based on inference from previously learned knowledge.

### The OFC$^{11/13}$-rmCD and OFC$^{11/13}$-MDm pathways differentially contribute to the sensitivity to past outcomes

Our results suggest that the two pathways contribute distinctly to different behavioral strategies. To further corroborate this dissociation, we performed a learning-model agnostic analysis in which we decomposed trials into specific events related to each behavioral strategy. First, we focused on experience-based updating during the NOVEL task. If the ability to update values based on past experience is impaired, the behavior following unpredicted positive experiences (i.e., obtaining a good result after choosing the previously unchosen option) should favor repeating the same choice, and vice versa following negative experiences (Fig. 4a). Consistent with the model-fitting analysis, silencing OFC$^{11/13}$ or the OFC$^{11/13}$-rmCD pathway significantly impaired performance following both positive and negative experiences (Fig. 4c, d), suggesting that their contribution to updating stimulus-reward associations is based on both positive and negative experiences. In contrast, silencing the OFC$^{11/13}$-MDm pathway only impaired performance following negative experiences (Fig. 4e), likely reflecting minor deficits at the immediate post-reversal period (Fig. 3d). Similarly, an asymmetric deficit was induced by silencing the OFC$^{11/13}$-MDm pathway during the FAMILIAR task (Supplementary Fig. 10).

Next, we focused on inference-based updating during the FAMILIAR task. We found that the monkeys could adapt to the new stimulus-reward association after a single experience of a reversal (1st trial, Fig. 4b, left), even when the subsequent trial did not include options that appeared in the previous trial (inference trial, Fig. 4b, right). Monkeys exhibited a greater percentage of optimal choices in the inference trial than in the 1st trial (inference vs. first trials; Mk#1: 72.2% vs. 19.6%; Mk#2: 59.3% vs. 17.7%) under the baseline control conditions, indicating that they inferred the stimulus-reward associations without needing direct experience. This inference-based behavioral adaptation was significantly impaired after silencing OFC$^{11/13}$ or the OFC$^{11/13}$-MDm pathway (Fig. 4f, h). In contrast, silencing the OFC$^{11/13}$-rmCD pathway had no effect on the inference trials (Fig. 4g). Taken together, these results support the conclusion that the OFC$^{11/13}$-rmCD pathway is essential for updating value via positive and negative experiences, whereas the OFC$^{11/13}$-MDm pathway is selectively involved in behavioral changes following negative experiences—a capacity that is critical for rapid behavioral adaptation using an inference-based strategy based on prior knowledge of the situation.

## Discussion

Here, by combining chemogenetic silencing of individual neural pathways with a model-fitting approach, we demonstrate that the two natural strategies for updating subjective value of external stimuli rely on two distinct neural pathways, each originating in the OFC$^{11/13}$ and projecting to a different subcortical brain region. Silencing the OFC$^{11/13}$-rmCD pathway impaired performance when monkeys updated option values through direct experience of both positive and negative changes of stimulus-reward associations, whereas silencing the OFC$^{11/13}$-MDm pathway impaired performance when monkeys updated the values based on inference that was guided by the negative experience that resulted from a change in the task context. This dissociable contribution of neural pathways provides new insights into the neural basis of value-based adaptive decision-making in primates.

In this study, we devised two behavioral tasks that allowed us to dissociate how stimulus-reward associations are updated: the NOVEL task for experience-based updating and the FAMILIAR task for inference-based updating. Indeed, our model-fitting analysis suggested that the monkeys solved these tasks using, primarily, the respective value-updating strategies proposed (Fig. 1). However, we had extensively trained the monkeys on the NOVEL task before the experimental sessions, which likely led to the development of a general understanding task structure, including the potential for reversal. Improved performance in the post-reversal phase, relative to the acquisition phase, suggests a cognitive ability that goes beyond purely making new experience-dependent associations from scratch. Therefore, we cannot exclude the possibility that some degree of inference was implemented during the NOVEL task, and deficits in both tasks induced by OFC silencing could have resulted from no longer being able to infer the changes in stimulus-reward associations following the reversals. However, the differential effects observed after silencing the OFC-rmCD pathway in the NOVEL task (specifically during the post-reversal phase), but not in the FAMILIAR task, led us to a more straightforward interpretation: the critical involvement of the OFC-rmCD pathway in the NOVEL task is specific to task context. Specifically, it appears to be essential for updating learned values based on experience following novel cue-reward associations. While this interpretation is straightforward, it does not rule out the possibility that some type of inference, such as awareness of potential reversal, could be a common element in both tasks.

The OFC has been thought to play an essential role in learning/updating stimulus-reward associations by integrating past experiences as consequences of our choices[2–6], although recent research has highlighted a specialized role in inference-based, or model-based,

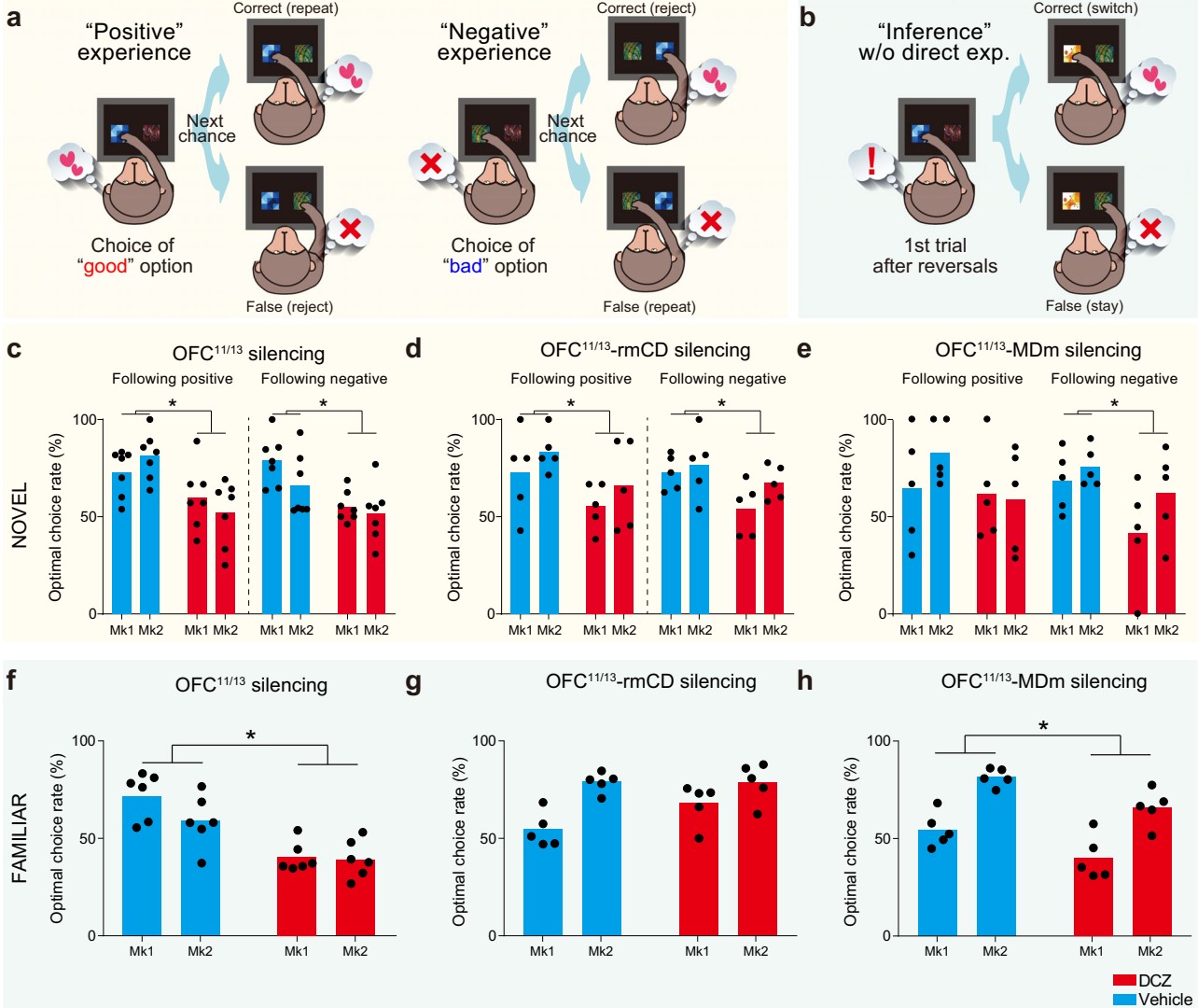

**Fig. 4 | Silencing of OFC[11/13]-rmCD and OFC[11/13]-MDm pathways differentially affected the sensitivity to past outcomes.** Schematic drawings showing the trials following positive (left) and negative (right) experience (**a**), and the "inference" trial following a single experience of the reversal of stimulus-reward associations (**b**) for the NOVEL and FAMILIAR tasks, respectively. Averaged optimal choice rate for trials following positive (left) and negative (right) outcomes for OFC[11/13] silencing (**c**) (two-way ANOVA, positive: treatment, $F_{(1,24)} = 14.8$, $p = 7.9 \times 10^{-4}$; subject, $F_{(1,24)} = 0.001$, $p = 0.97$; interaction, $F_{(1,24)} = 2.2$, $p = 0.15$; negative: treatment, $F_{(1,24)} = 14.3$, $p = 9.1 \times 10^{-4}$; subject, $F_{(1,24)} = 2.7$, $p = 0.11$; interaction, $F_{(1,24)} = 0.94$, $p = 0.34$), OFC[11/13]-rmCD silencing (**d**) (positive: treatment, $F_{(1,16)} = 4.8$, $p = 4.3 \times 10^{-2}$; subject, $F_{(1,16)} = 1.8$, $p = 0.20$; interaction, $F_{(1,16)} = 0.001$, $p = 0.97$; negative: treatment, $F_{(1,16)} = 6.0$, $p = 2.6 \times 10^{-2}$; subject, $F_{(1,16)} = 2.4$, $p = 0.14$; interaction, $F_{(1,16)} = 0.65$, $p = 0.43$), and OFC[11/13]-MDm silencing (**e**)

(positive: treatment, $F_{(1,16)} = 1.5$, $p = 0.23$; subject, $F_{(1,16)} = 0.51$, $p = 0.48$; interaction, $F_{(1,16)} = 0.90$, $p = 0.36$; negative: treatment, $F_{(1,16)} = 5.2$, $p = 3.7 \times 10^{-2}$; subject, $F_{(1,16)} = 2.4$, $p = 0.14$; interaction, $F_{(1,16)} = 0.56$, $p = 0.46$). Averaged optimal choice rate for inference trials after OFC[11/13] silencing (**f**) (two-way ANOVA, treatment, $F_{(1,20)} = 33.1$, $p = 1.3 \times 10^{-5}$; subject, $F_{(1,20)} = 2.5$, $p = 0.13$; interaction, $F_{(1,20)} = 1.7$, $p = 0.21$), OFC[11/13]-rmCD silencing (**g**) (treatment, $F_{(1,16)} = 2.7$, $p = 0.12$; subject, $F_{(1,16)} = 19.6$, $p = 4.2 \times 10^{-4}$; interaction, $F_{(1,16)} = 2.9$, $p = 0.11$), and OFC[11/13]-MDm silencing (**h**) (treatment, $F_{(1,16)} = 14.2$, $p = 1.7 \times 10^{-3}$; subject, $F_{(1,16)} = 43.7$, $p = 6.0 \times 10^{-6}$; interaction, $F_{(1,16)} = 0.021$, $p = 0.88$). For the OFC[11/13] silencing (**c**, **f**), data were obtained from $N = 7$ and 6 sessions for each treatment in each monkey for the NOVEL (**c**) and FAMILIAR (**f**) tasks, respectively. For silencing of each pathway (**d**, **e**, **g**, **h**), data were obtained from $N = 5$ sessions for each treatment, each task, and each monkey.

learning[30,31]. Recent reports suggest that rodent OFC, or the ventral PFC in humans, is not simply engaged in either experience- or inference-based value-updating strategies, but rather in regulating the switch between them according to current contexts[10,32]. Our present results are consistent with and extend this idea; we can now assign distinct roles for at least two different OFC projections to subcortical structures (the rmCD and MDm). Two regions downstream from the PFC—the striatum and MD thalamus—have been shown to be involved in value-updating. Many studies have suggested the involvement of the striatum in both experience- and inference-based updating, with a gradation within the striatum[15–17,33–35]. Importantly, a pathway-selective manipulation study in rodents demonstrated that selective ablation of the OFC-accumbens pathway induced impaired performance

following a reversal that was similar to what we show here, with a slight difference in the sensitivity to positive outcomes[36]. The MD thalamus has also been suggested to be involved in inference-based updating[18,37], or in adapting to shifts in task context, not simply in changes to stimulus-outcome contingencies[38]. The critical contribution of the interactions between the PFC and the MD thalamus for adaptation to shifts in task context has also been demonstrated in rodents[39,40]. These previous reports imply that OFC sends information necessary for implementing each type of value update, e.g., reward prediction and prediction error signal[41–43] or task context[44], to rmCD and MDm, which should be identified in future studies.

The OFC is widely thought to play a central role in updating and maintaining information about possible outcomes[6,11]. Both lesion and

recording studies have shown that the OFC is critical for updating associations between stimuli and outcomes, but not for acquiring such associations[20,45], which was also the case in our study (Fig. 2). Previous research has also suggested that the OFC is required for reversal learning in various species[46–48]. However, recent studies have challenged this view by showing that selective lesion/inactivation of the OFC in macaque monkeys had no effect in a simple reversal learning paradigm[25,49], a result consistent with our observations (Supplementary Fig. 4). Recently, another study suggested that the OFC is essential for updating the desirability (i.e., quality or quantity) of reward-associated stimuli, but not their availability (i.e., the probability of receiving rewards)[20]. Our current results, in which OFC silencing severely impaired performance on a behavioral task that requires monkeys to associate multiple stimuli with multiple reward amounts and to update their associations, support this view. Notably, OFC silencing did not impact performance on the Wisconsin Card Sorting Task (WCST, Supplementary Fig. 5), which is commonly used to assess behavioral flexibility. Although previous lesion studies in monkeys have reported deficits in WCST performance following aspirating lesions[26], subsequent studies have suggested that these deficits may have been due to damaged fibers that pass near the OFC[25]. Because outcomes on the WCST are typically binary, either receiving a reward or not, our results suggest that primate OFC, at least the area that we focused on in this study (Brodmann's area 11/13), is specialized for representing stimulus-reward associations that are based on the desirability of possible outcomes.

Importantly, the two pathways we focused on form part of a broader circuit for value-updating. For example, the amygdala is known to be involved in value-based behavior through interactions with the OFC. It has been shown that lesion of the monkey amygdala affects value representation in OFC neurons[50,51], and that crossed lesions of the OFC and amygdala interrupt reinforcer devaluation[52,53], suggesting that their interactions are important for inference-based value updating. Lesions of the amygdala were also reported to result in inference-based enhancements due to reductions in the amount of evidence suggesting a reversal has occurred, which causes frequent switching after a negative outcome[54,55]. However, we did not examine the OFC-amygdala pathway because positive PET signals were not detected in the amygdala (Supplementary Fig. 1), which is consistent with previous anatomical studies indicating relatively weak direct projections from OFC[11/13] to this area[56,57]. Even without any direct OFC-to-amygdala connection, we must consider the fact that these brain regions may interact through other routes, such as direct projections from the amygdala to the OFC[58] or indirect connections via midbrain dopamine neurons[59,60]. Future studies should aim to investigate how the OFC and the amygdala interact and contribute to value updating.

In the current study, we were able to dissect the functions of the OFC-subcortical circuits by using the imaging-guided pathway-selective synaptic silencing technique that we recently developed[24]. In that study, we also demonstrated a functional double dissociation in the pathways from the dorsolateral prefrontal cortex (dlPFC) to the CD and MD thalamus. Given the parallel forms of cortico-basal ganglia-thalamo-cortical circuits[61], these consistent observations lead us to predict that other prefronto-striatal and prefronto-thalamus pathways, e.g., pathways from the ventrolateral PFC to the caudate and MD thalamus, may also have specialized roles aligned with the cognitive functions associated with their prefrontal origin, as recently investigated in the rodents[62]. On the other hand, convergence of projections from different prefrontal areas to specific striatal zones has also been reported[63]. Indeed, the PET data from the current study and our previous research[24], indicate that a part of the rmCD receives projections from both the dorsolateral and orbital prefrontal cortex, seemingly corresponding to a previously identified "convergence zone"[63]. These observations suggest that this area may be involved in the integration

of information from different cortico-basal ganglia-thalamo-cortical circuits. Future studies should aim to better understand the overall architecture of these networks by identifying where and how they communicate with each other and how the information processed in each network is integrated.

The current study had some limitations. First, our data were obtained from only two monkeys, fewer than the norm for behavioral ablation studies. Nevertheless, the strength of our conclusions about the functional roles of the OFC and its projections to the two subcortical regions is supported by statistically significant and consistent findings across multiple experiments in these two monkeys (for a few exceptions, please see each figure legend). Our chemogenetic approach allows for a rigorous within-subject design—in contrast to ablation techniques that generally require larger sample sizes (typically three or more subjects) due to individual variability—to study the effect of region/pathway manipulations in the same individuals. Second, although our results demonstrated that selective silencing of either pathway induced robust effects comparable to those observed by OFC[11/13] silencing per se, especially in silencing the OFC[11/13]-rmCD pathway in the NOVEL task, this does not exclude the possibility that other regions/pathways also play important roles in generating these behaviors. Furthermore, we cannot determine whether we were able to inhibit an entire subset of projections from the OFC[11/13]—as our methods rely on the diffusion of agonist solution and proportion of neurons that expressed DREADDs—only that it was sufficient to drive selective behavioral deficits.

In summary, leveraging the technical advantage of imaging-guided pathway-selective chemogenetic silencing, we have demonstrated the dissociable contributions of two OFC-subcortical pathways to different value-updating strategies. The identification of causal relationships between a specific neural pathway and a cognitive function, as demonstrated in this study, can complement what we have learned from human studies, but which cannot be directly tested in humans for ethical reasons. Thus, in addition to providing new insights into the neural basis of value-based adaptive decision-making in non-human primates, our findings have potential implications for understanding certain psychiatric disorders, such as obsessive-compulsive disorder (OCD). Our findings are particularly relevant to OCD because abnormal functional connectivity between the OFC and the caudate nucleus is frequently reported in patients with OCD[64].

## Methods

### Subjects

Two experimentally naïve male Japanese monkeys (*Macaca fuscata*) participated in the experiments (Mk#1: 7.2 kg; Mk#2: 6.8 kg; both aged 4 years at the beginning of experiments). The monkeys were kept in individual primate cages in an air-conditioned room. A standard diet, supplementary fruits/vegetables, and a tablet of vitamin C (200 mg) were provided daily. All experimental procedures involving the monkeys were carried out in accordance with the Guide for the Care and Use of Nonhuman primates in Neuroscience Research (The Japan Neuroscience Society; https://www.jnss.org/en/animal_primates) and were approved by the Animal Ethics Committee of the National Institutes for Quantum Science and Technology.

### Viral vector production

Mk#1 was co-injected with two AAV vectors, one expressing hM4Di and the other expressing GFP (AAV2-CMV-hM4Di and AAV2-CMV-AcGFP; $2.3 \times 10^{13}$ and $4.6 \times 10^{12}$ particles/mL, respectively). Mk#2 was injected with an AAV vector expressing both hM4Di and GFP (AAV2.1-CaMKII-hM4Di-IRES-AcGFP, $1.0 \times 10^{13}$ particles/mL). AAV vectors were produced by a helper-free triple transfection procedure, and were purified by affinity chromatography (GE Healthcare, Chicago, USA). Viral titer was determined by quantitative PCR using Taq-Man technology (Life Technologies, Waltham, USA)[65].

## Surgical procedures and viral vector injections

Surgeries were performed under aseptic conditions in a fully equipped operating suite. We monitored body temperature, heart rate, SpO$_2$, and tidal CO$_2$ throughout all surgical procedures. Monkeys were immobilized by intramuscular (i.m.) injection of ketamine (5–10 mg/kg) and xylazine (0.2–0.5 mg/kg) and intubated with an endotracheal tube. Anesthesia was maintained with isoflurane (1–3%, to effect). Before surgery, magnetic resonance (MR) imaging (7 tesla 400 mm/SS system, NIRS/KOBELCO/Brucker) and X-ray computed tomography (CT) scans (Accuitomo170, J. MORITA CO., Kyoto, Japan) were performed under anesthesia (continuous intravenous infusion of propofol 0.2–0.6 mg/kg/min). Overlay MR and CT images were created using PMOD® image-analysis software (PMOD Technologies Ltd, Zurich, Switzerland) to estimate stereotaxic coordinates of target brain structures. After surgery, prophylactic antibiotics and analgesics (cefmetazole, 25–50 mg/kg; ketoprofen, 1–2 mg/kg) were administered.

The bilateral OFCs (BA11 & BA13) of each monkey were injected with the AAV vectors (Fig. 1a). The injections were performed under direct vision using the same types of surgical procedures as in a previous study[66]. Briefly, after retracting skin, galea, and muscle, the frontal cortex was exposed by removing a bone flap and reflecting the dura mater. Then, handheld injections were made under visual guidance through an operating microscope (Leica M220, Leica Microsystems GmbH, Wetzlar, Germany), with care taken to place the beveled tip of a microsyringe (Model 1701RN, Hamilton) containing the viral vector at an angle oblique to the brain surface. The needle (26 Gauge, PT2) was inserted into the intended area of injection by one person and a second person pressed the plunger to expel approximately 1 μL per penetration. For Mk#1, total volumes of 54 μL and 50 μL were injected via 54 and 50 tracks in the left and right hemispheres, respectively. For Mk#2, total volumes of 53 μL and 49 μL were injected via 50 and 47 tracks in the left and right hemispheres, respectively.

## PET imaging

PET imaging was conducted as previously reported[24]. Briefly, PET scans were conducted before injection of vectors and at 45 days after injection for both monkeys. PET scans were performed using a microPET Focus 220 scanner (Siemens Medical Solutions USA, Malvern, USA). Monkeys were immobilized by ketamine (5–10 mg/kg) and xylazine (0.2–0.5 mg/kg) and then maintained under anesthetized condition with isoflurane (1%–3%) during all PET procedures. Transmission scans were performed for approximately 20 min with a Ge-68 source. Emission scans were acquired in 3D list mode with an energy window of 350–750 keV after intravenous bolus injection of [$^{11}$C]clozapine (for Mk#1; 375.5–394.7 MBq) or [$^{11}$C]DCZ (for MK#2; 324.9–382.3 MBq). Emission data acquisition lasted 90 min. To estimate the specific binding of [$^{11}$C]DCZ in Mk#2, regional binding potential relative to nondisplaceable radioligand (BP$_{ND}$) was calculated by PMOD® with an original multilinear reference tissue model (MRTMo). To visualize the expression of DREADDs, contrast (subtraction) of images taken before and 45 days after vector injection were created using PMOD for SUV (standardized uptake value) for Mk#1 and BP$_{ND}$ for Mk#2 by investigating whether differential PET signals were observed at the target sites.

## Drug administration

DCZ (HY-42110; MedChemExpress) was dissolved in dimethyl sulfoxide (DMSO, FUJIFILM Wako Pure Chemical Co.), aliquoted and stored at −30 °C. For systemic intramuscular injection, this stock solution was first diluted in saline to a final volume of 1 mL (2.5% DMSO in saline), and thus achieving a dose of 100 μg/kg. DCZ solution was injected 15 min before the beginning of the experiments, unless otherwise noted. Fresh solution was prepared on each day of usage. We performed at most one injection experiment per week for each treatment (vehicle and DCZ), repeating the NOVEL and FAMILIAR tasks seven and six times, respectively.

For microinfusion, DCZ was first dissolved in DMSO and then diluted in PBS to a final concentration of 100 nM. We prepared fresh solutions on the day of usage. We used two stainless steel infusion cannulae (outer diameter 300 μm; Muromachi-Kikai) inserted into each target region: rmCD and MDm, and ventral putamen for additional experiments (Fig. S7). Each cannula was connected to a 10-μL microsyringe (#7105KH; Hamilton) via polyethylene tubing. These cannulae were advanced via guide tube by means of an oil-drive micromanipulator. DCZ solution or PBS was injected at a rate of 0.25 μL/min by auto-injector (Legato210; KD Scientific) for a total volume of 3 μL for each hemisphere. The injection volumes were determined based on a previous study reporting that injections of 3 μL and 1.5 μL resulted in a diameter of aqueous spread in the monkey brain of approximately 5–6 mm and 3–4 mm, respectively[67]. We chose sufficient volumes to cover the hM4Di-positive terminal sites, which had diameters of 5–7 mm and 3–4 mm for the rmCD and MDm, respectively, as measured by increased PET signals. Because the MDm is located close to the midline, we placed the canulae laterally near the MDm so that the injected solution would diffuse into the entirety of the MDm (Fig. 3b). CT image was obtained to visualize the infusion cannulae in relation to the chambers and skull following each infusion. The CT image was overlaid on MR and PET images obtained beforehand using PMOD to verify that the infusion sites (tips of the infusion cannulae) were located in the target (presumed hM4Di-positive terminal regions identified as increased PET signals). The behavioral session began approximately 30 min after the end of the infusion and lasted approximately one hour in both tasks, thus the sessions ended 90 min after the end of the infusions. We performed at most one infusion in one are per week for one area, and repeated five times for each area. The order of the experiments for each monkey, including systemic injections and local infusions into each area, is summarized in Supplementary Fig. 8.

## Behavioral tasks

The monkeys were tested with two versions of modified reversal learning tasks in which they were required to choose either of two visual stimuli (out of a set of five) presented on a computer screen. Behavioral testing was conducted in a sound-attenuated room. The monkeys sat on a monkey chair from which they could reach out one with hand to touch an LCD display placed in front of them. The behavioral task was controlled by a computer using commercially available software (Inquisit, Millisecond). A monkey initiated a trial by touching a sensor mounted on the chair, which caused a small white circle to appear in the center of the display. After a delay of 0.5 s, the circle disappeared and two stimuli of the five possible stimuli were presented simultaneously on the left and right side of the display. If the monkey touched either stimulus, it could receive a reward from the spout placed in front of its mouth. If the monkey released the touch lever before the presentation of visual stimuli, the trial was aborted and repeated after a 3–4 s inter-trial interval. Each stimulus was associated with 1, 2, 3, 4, or 5 drops of juice.

In the NOVEL task, a new set of visual stimuli was introduced each session, which required the monkeys to learn a new set of stimulus-reward associations. A daily session consisted of 90 acquisition-phase trials, followed by the reversal of the stimulus-reward associations, and then 210 post-reversal trials. The reversal was conducted such that a stimulus previously associated with 1 drop of juice became associated with 5 drops of juice, and one associated with 2 drops became associated with 4 drops, and vice versa. The combination of visual stimuli seen on each trial was pre-determined pseudorandomly so that each combination appeared once every 10 trials in a round-robin fashion.

In the FAMILIAR task, a fixed sets of five visual stimuli were used throughout the experiments. This ensured that the monkeys became familiar with all possible stimulus-reward associations for the two sets before and after reversals. If the optimal choice rate (i.e., the proportion of trials in which the option associated with the greater reward was

chosen) in 30 consecutive trials passed 76%, the associations were reversed. Daily sessions comprised 300 and 400 trials for Mk#1 and Mk#2, respectively. Both monkeys were trained on the NOVEL task and then the FAMILIAR task. As a control, the monkeys were tested on a simple reversal learning task (Supplementary Fig. 4) in which only two novel stimuli were introduced in each session. The stimuli were associated with 5 drops of juice reward or no reward.

One monkey (Mk#2) was also tested on a reinforcer devaluation task and on a modified version of the Wisconsin Card Sorting task, as previously described[25,26]. Briefly, in the reinforcer devaluation task, the monkey was required to choose one of two objects that were placed above two holes located in a wooden plate. For one set of objects, food 1 (peanut) was delivered when the object was selected, whereas for the other set of objects, food 2 (raisin) was delivered. The associations between the objects and reward type were fixed throughout training. The monkey was tested on 4 consecutive days in a week: review of object-reward associations (Day 1), baseline choice test (Day 2), review of object-reward associations (Day 3), and choice test following selective satiation (devaluation of a food) with vehicle or DCZ administration 15 min before devaluation (Day 4). For devaluation, the monkey was given up to 30 min to consume as much of either food as it wanted. The devaluation procedure was deemed to be complete when the monkey refrained from retrieving food from the food chamber for 5 min. The monkey's ability to adaptively shift away from choosing objects associated with the devalued food was calculated as the "proportion shifted" as below,

$$\text{Proportion shifted} = \frac{(F1N - F1D) + (F2N - F2D)}{(F1N + F2N)} \cdots \quad (1)$$

where F1 and F2 represent choices associated with the two food types (peanut and raisin) in each week in which that food type was devalued, and D and N respectively represent the data for devaluation (Day 4) and baseline (Day 2). A version of the Wisconsin Card Sorting Test, which was modified for monkeys to comprise only two rules (color and shape), was used as previously reported[26]. Briefly, in each trial, the monkey initiated a trial by touching a sensor mounted on the chair, which caused a small white circle to appear in the center of the display. After a delay of 0.5 s, the circle disappeared and a sample stimulus appeared in the center of the display, followed after 1 s by horizontal (left, center, right) presentation of three test stimuli below the sample stimulus, one matching the sample stimulus in color, another matching in shape, and the third not matching in either color or shape. The sample and test stimuli were randomly chosen from a set of colored shapes (colors = red, yellow, green, and blue; shapes = triangle, circle, square, and cross). The monkeys had to touch the test stimulus that matched the sample in color or in shape within 2000 ms. The monkey was given a reward (drops of juice) after a correct target selection, and the sample and test items were extinguished upon reward delivery. If the animal chose an incorrect option, reward was not given and a white square around the chosen stimulus appeared for 1 s as a visual feedback for error. When the monkey reached 85% correct performance in 20 consecutive trials or 90% correct in 10 consecutive trials, the relevant rule (matching by color or matching by shape) was changed without notice to the monkey. A minimum of 20 trials was provided in each block even if the monkey reached the criterion earlier. Each daily session consisted of 400 trials and the first rule of the day alternated between days.

## Statistics

Throughout the manuscript, we compare optimal choice rate and learning rate between conditions. For analysis of optimal choice rate, we divided the session into three phases. For the NOVEL task, the phases were as follows: pre-reversal (Pre; the 90 trials before the reversal), early phase of the post-reversal (PoE; the first 100 trials after the reversal), and late phase of the post-reversal (PoL; the next 110

trials after reversal). For the FAMILIAR task, they were Pre (10 trials before the reversal), PoE (the first 5 trials after reversal), PoL (the next 5 trials after reversal). Then, we first conducted a three-way ANOVA (subject × phase × treatment) to determine whether there was a significant difference in treatment (Vehicle vs. DCZ) or any interaction between phase and treatment, indicating deficits in a specific phase(s). If there were, we subsequently conducted two-way ANOVAs (subject × treatment) for each phase to determine in which phase(s) any significant differences in treatment occurred. For analysis of learning rate, comparisons were analyzed with two-way repeated-measures ANOVA (subject × treatment) to find the effect of each treatment and any individual differences. The analyses were conducted using GraphPad Prism 9. For the NOVEL task, the optimal choice rate was averaged across 10 trials of pseudo-random stimulus-reward combinations (see "Behavioral task") for each session. In the FAMILIAR task, the optimal choice rate around the time of reversal was averaged across each reversal for each session. Reaction time was defined time between releasing the bar and touching the object on the screen, and data were averaged across 10 trials as described above.

For a learning-model that incorporates an agnostic analysis of the ability to update values based on past experience (Fig. 4), we extracted the trials in the PoE phase following unpredicted positive and negative experiences as follows. An "unpredicted positive" experience was defined as a single choice of higher valued options (4 or 5 drops) following a non-choice of that higher valued option in a preceding trial, and vice-versa (lower valued options, 1 or 2 drops) for negative experiences. Then, the averaged optimal choice rate in the next trials that included that option following these unexpected outcomes was calculated. A concrete example for a positive-experience would be as follows: the monkey is presented with "5-drops-of-juice" and "2-drops-of-juice" options, but does not choose the 5-drops option. Subsequently, it is presented with the choice between 3-drops and 5-drops stimuli and chooses the 5-drops option (this is the unpredicted positive experience). A future trial in which the monkey has to choose between the 1-drop and 5-drops options would be the one extracted for analysis. Similarly, for a negative experience: the monkey is presented with a choice between 1-drop and 2-drops stimuli, and does not choose the 1-drop option. Subsequently, when presented with 1-drop and 4-drop stimuli, it chooses the 1-drop option (this is the unexpected negative experience). A future trial in which the monkey has to choose between 1-drop and 3-drop stimuli would be extracted for analysis.

## Histology and immunostaining

For histological inspection, monkeys were deeply anesthetized with an overdose of sodium pentobarbital (80 mg/kg, i.v.) and transcardially perfused with saline at 4 °C, followed by 4% paraformaldehyde in 0.1 M phosphate buffered saline (PBS), pH 7.4. The brain was removed from the skull, postfixed in the same fresh fixative overnight, saturated with 30% sucrose in phosphate buffer (PB) at 4 °C, and then cut serially into 50-μm-thick sections with a freezing microtome. For visualization of immunoreactive GFP signals (co-expressed with hM4Di), a series of every 6th section was immersed in 1% skim milk for 1 h at room temperature and incubated overnight at 4 °C with rabbit anti-GFP monoclonal antibody (1:500, G10362, Thermo Fisher Scientific) in PBS containing 0.1% Triton X-100 and 1% normal donkey serum (S30-100ML, Sigma-Aldrich) for 2 days at 4 °C. The sections were then incubated in the same fresh medium containing biotinylated donkey anti-rabbit IgG antibody (1:1,000; Jackson ImmunoResearch, West Grove, PA, USA) for 2 h at room temperature, followed by avidin-biotin-peroxidase complex (ABC Elite, Vector Laboratories, Burlingame, CA, USA) for 2 h at room temperature. For visualizing the antigen, the sections were reacted in 0.05 M Tris-HCl buffer (pH 7.6) containing 0.04% diaminobenzidine (DAB), 0.04% NiCl2, and 0.003% $H_2O_2$. The sections were mounted on gelatin-coated glass slides, air-dried, and cover-slipped. A portion of the other sections was Nissl-

stained with 1% Cresyl violet. Images of sections were digitally captured using an optical microscope equipped with a high-grade charge-coupled device (CCD) camera (Biorevo, Keyence, Osaka, Japan).

## Model-fitting analysis

We constructed two mathematical models termed "EXP" and "INF" to investigate whether monkey behavior could be explained by experience- or inference-based strategies.

In the EXP model, the probability that the learner (i.e., monkey) chooses stimulus $S_i$ when stimuli $S_i$ and $S_j$ are presented ($i,j = 1, \cdots, 5; i \neq j$) is given by

$$Pr[S = S_i] = \frac{exp(\beta V_i)}{exp(V_i) + exp(\beta V_j)} \cdots \quad (2)$$

where $\beta$ is a parameter controlling the exploration-exploitation trade-off (inverse temperature), and $V_i$ is the subjective value of stimulus $S_i$, which is interpreted as the amount of reward expected by the learner after choosing $S_i$. $V_i$ is updated after each trial based on the Rescorla–Wagner rule, which applies the stochastic gradient-descent algorithm to minimize the squared error between the expected reward for the chosen stimulus and the experienced reward. The update rule after choosing stimulus $S_i$ is given by

$$V_i \leftarrow V_i - \alpha(V_i - r_{obs}) \cdots \quad (3)$$

where $\alpha$ is the learning rate and $r_{obs}$ is the amount of the reward provided in the current trial. All $V_i$ ($i = 1, \cdots, 5$) were assumed to be zero at the beginning of each session, and the learning rate and inverse temperature were fitted to the behavioral data in each session via maximum likelihood estimation.

In the INF model, we assume that the learner knows that the stimulus-reward association takes one of the two possible patterns. Taking this assumption, we prepared two sets of $V_i$:

$$V_i^A = i \, (i = 1, \cdots, 5) \cdots \quad (4)$$

for pattern A, and

$$V_i^B = 6 - i \, (i = 1, \cdots, 5) \cdots \quad (5)$$

for pattern B. In the INF model, the learner's choice is assumed to follow a mixture of the two softmax distributions corresponding to the two possible stimulus-reward association patterns. The probability that stimulus $S_i$ is chosen when stimuli $S_i$ and $S_j$ are presented is given by

$$Pr[S = S_i] = w\left\{\frac{exp(\beta V_i^A)}{exp(\beta V_i^A) + exp(\beta V_j^A)}\right\} + (1-w)\left\{\frac{exp(\beta V_i^B)}{exp(\beta V_i^B) + exp(\beta V_j^B)}\right\} \cdots \quad (6)$$

where $\beta$ is the inverse temperature and $w$ is a weight parameter adaptively learned based on the experienced rewards. If $w = 1$, the learner's choice is assumed to follow the softmax distribution corresponding to stimulus-reward association pattern A, and if $w = 0$, the choice is assumed to follow the softmax distribution corresponding to stimulus-reward association pattern B. Thus, this weight parameter can be interpreted as a parameter representing the confidence that the current stimulus-reward association is pattern A out of the two possible patterns. Unlike in the EXP model, in the INF model, the inputs to the softmax function are fixed, and the parameter weighting the two softmax distributions is updated after each trial. Like the EXP model, the weight parameter is updated by the update rule derived from the stochastic gradient-descent algorithm, which minimizes the squared error between the expected reward for the

chosen stimulus and the experienced reward. The update rule after choosing stimulus $S_i$ is given by

$$w \leftarrow w - \alpha\left\{\left(wV_i^A + (1-w)V_i^B\right) - r_{obs}\right\}\left(V_i^A - V_i^B\right) \cdots \quad (7)$$

where $\alpha$ is the learning rate and $r_{obs}$ is the observed reward in the current trial. $w$ is assumed to be 0.5 at the beginning of each session and is assumed to take a value in the range of [0.0 1.0] at each update by replacing its value with 0.0/1.0 if it is smaller/larger than 0.0/1.0. As in the EXP model, the learning rate and inverse temperature were fitted to the behavioral data in each session via maximum likelihood estimation.

Because we were interested in the updating behavior during the post-reversal period, we modified the two models described above for the current study. In the modified models, the trials in each session were divided into two groups: 1) the trials in the post-reversal period, and 2) others. The post-reversal period consisted of all trials after the reversal in the NOVEL task, and the five trials after each reversal in the FAMILIAR task. While the learning rate in the basic models is fixed in each session, in the modified models we assumed that it took a different value each of the two trial groups. The learning rate for the post-reversal period is considered to better reflect the updating behavior, and a total of three parameters (two learning rates and inverse temperature) were fitted to the given behavioral data. Unless stated otherwise, the learning rates for the post-reversal periods derived from the modified models were reported in this paper.

To compare the ability of the EXP and INF models to explain the behavioral data, we performed Bayesian model comparison. Following the procedure in previous studies[68,69], the Bayesian information criterion (BIC) was computed for each model and each session, then the Bayesian posterior probability for each model was computed based on those BIC values.

## Reporting summary

Further information on research design is available in the Nature Portfolio Reporting Summary linked to this article.

## Data availability

The Source data are provided as a Source Data file at https://github.com/minamimoto-lab/2023-Oyama-OFC. Source data are provided with this paper.

## Code availability

The Python code used for the analysis is available at https://github.com/minamimoto-lab/2023-Oyama-OFC.

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

## Acknowledgements

This study was supported by MEXT/JSPS KAKENHI Grant Numbers JP18K15353, JP21K07268, JP22H05521 (to K.O.), and, JP17H02219 (to T.H.), JP22H05157 (to K.I.), JP19H05467 (to M.T.), and JP15K21742, JP20H05955 (to T.M.), by JST PRESTO Grant Numbers JPMJPR22S3 (to K.O.), and JPMJPR2128 (to KMajima) and by AMED Grant Numbers JP23dm0307007 (to T.H.), JP21dm0107146 (to T.M.), JP20dm0307021 (to K.I.), and JP21dm0207077 (to M.T.). Two Japanese monkeys were provided by the Japan MEXT National Bio-Resource Project "Japanese Monkeys". We thank Drs. Brian Russ and Junya Hirokawa for their valuable comments on an earlier version of the manuscript, and Jun Kamei, Ryuji Yamaguchi, Yuichi Matsuda, Yoshio Sugii, Takashi Okauchi, Rie Yoshida, Maki Fujiwara, and Mayuko Nakano for their technical assistance. We also thank Katsushi Kumata and Dr. Ming-Rong Zhang for producing the radioligand.

## Author contributions

Conceptualization: T.M.; investigation: K.O., Yukiko.Hori., and Y.N.; Model construction: K.Majima; Resources: K.I. and M.T.; Surgery: M.E. and R.S.; Visualization: K.O.; Writing – review & editing: K.O., K. Majima, Y.N., Yukiko.Hori, T.H., M.E., K. Mimura., N.M., A.F., Yuki Hori, H.I., K.I., R.S., M.T., N.Y., M.H., B.R., and T.H.

## Competing interests

The authors declare no competing interests.
