## [Peer Review File · Nature Communications]

Distinct roles of monkey OFC-subcortical pathways in adaptive behaviorREVIEWER COMMENTS

Reviewer #1 (Remarks to the Author):

In this article by Oyama et al., the authors applied their state-of-art imaging guided pathway-selective manipulation technique with DREADD and dissociated the functions of OFC-rmCD pathway and OFC-MDm pathway; that is, the former pathway regulates the experience-based value updating, while the latter pathway regulates inference-based value updating by using namely the NOVEL task and FAMILIAR task. The authors also fitted the behavioral data with two types of reinforcement learning models and empowered this hypothesis. This study is further extension of their 2021 paper on functional dissociation of dlPFC-CD and dlPFC-MD pathways, which also dissociated the functions of the two pathways originating from the same cortical regions. The experiments were carefully designed and results were clear and striking. I celebrate the authors for their achievement but have some conceptual comments on the manuscript.

Major comment;

The authors claimed that the NOVEL task is related to the experience-based value updating and FAMILIAR task is related to the inference-based value updating. But to me, there seems to be more than that. In the case of NOVEL task, DCZ administration did not affect the acquisition phase but suppressed the reversal phase. If the subject is completely naïve to the stimulus sets and the rule, the reversal should also be relearning of the association of stimulus and reward from the beginning. If so, DCZ effect should be similar (no effect). But the present results indicate that the acquisition and reversal are different process, that is, the subject should have known the rule that the pattern (1,2,3,4,5) should be reversed to the pattern (5,4,3,2,1). This means there should be a process of strong inference in this NOVEL task, too. Thus, I doubt that the NAÏVE task is purely experience-dependent, free of inference. To me, it seems that the FAMILIAR task implements the low level (or more direct) inference, and the NOVEL task might implement the higher-order inference.

Minor comments;

1. Figure 4 does not include the inset indicating what is red and what is blue.
2. In the legend of Extended Figure 1, there is no explanation of abbreviations PUT and AMG.
3. In the method (L646), the authors described the behavioral session began 30 min after the end of infusion. In the current experiments, the DCZ should have been diffused to the areas of larger volume (diameters of 5-7 mm or 3-4 mm). I wonder whether the effects were stable during the recording time. The authors should describe more about the recording time after infusion and how stable the effects were during the session.

Reviewer #2 (Remarks to the Author):

This study investigates the contribution of three pathways from OFC in experience-based and inference-based updating of stimulus-reward associations. Two monkeys were trained on two versions of a stimulus-reward learning and reversal task using either novel or familiar stimuli. DREADDs were then used to inactivate areas 11 and 13 as well as their separate pathways to the caudate, putamen and MD thalamus. OFC inactivation disrupted performance on both tasks whilst the pathway from OFC to MD disrupted the inference-based and not the experience-based task whilst the projection to the medial Caudate produced the opposite effect. Projections to the putamen had no effect on the novel task and so was not explored any further.

The results are novel and of potential interest to the scientific community. However, the major drawback is that only two monkeys were used and the overall flavour of this study is that it was a pilot. Two is a very low 'n' for such intervention studies, even in non-human primates, and would usually be considered a pilot study, even more so since an important control manipulation, injecting the activator in the absence of DREADDs was only performed in one monkey, as was the use of the WCST to rule out effects of OFC on high cognitive load. Moreover, the two monkeys received different viruses, one in which a single virus expressed both fluorescence and hM4Di and the other, in which the fluorescence was an independent virus. Given the concerns of replication failures across scientific studies it is unclear why only two monkeys were used in this intervention study? Whilst each intervention on each task was repeated on 5-7 occasions in each monkey this nevertheless does not help show the consistency of effects across monkeys, a very important consideration. Related to this is the fact that it is unclear whether these animals were experimentally naïve before the start of the study as past history can impact how an animal may learn behavioural tasks. Thus, overall, the results of this study need replicating in additional monkeys, preferably using the same viral vectors.

With regards the conclusions from this study, it would have been nice if there had been some discussion regarding the results in the context of the different neural pathways that have been implicated. Why for example would experience-based updating be dependent upon the caudate pathway whilst inference-based updating be more dependent on the MD thalamus pathway? What else is known about the role of the striatum and thalamus in such cognitive functions that may explain this differentiation? Moreover, it is stated in the final sentence that these findings are relevant to our understanding of neuropsychiatric disorders but why? Is there evidence of such specific, dissociable deficits in patients? It is also stated in the discussion that: "Although the OFC and dlPFC are located in different parallel cortico-basal ganglia-thalamo-cortical circuits, the observed similarity that different subcortical pathways have different functions suggests that the parallel prefrontal-subcortical networks might share a common neural basis for adaptive behavior". Just because multiple pathways from distinct prefrontal regions e.g. dlPFC and OFC have been shown to perform different functions it is premature to suggest that they share a common neural basis for adaptive behaviour? In many ways it might be surprising if different pathways from the

same region did perform the same function? Different pathways from many other cortical and subcortical regions have been shown to perform different functions so I don't think the finding says anything special about dlPFC and OFC and the parallel cortico-striatal pathways they contribute to?

Specific questions are laid out below:

Results.

Describing this region as lateral OFC is confusing as very often the neighbouring area 12/47 is very often called lateral OFC too. So, I recommend just referring to this area as OFC11-13 so as to avoid confusion.

In the figures it would be preferable to see the data for each monkey separately rather than averaging the two monkeys performance on repeated sessions. In Fig 1 the two monkeys could be depicted on the same graphs in 1b and c.

Not clear what 'latter' is referring to on line 121. Page 6.

For completeness/thoroughness, in Fig 2, can the bilateral PET signal in OFC be shown, rather than just a unilateral image please? Since only two monkeys took part in this study then it is important to show both monkey's data.

Can an explicit statement be made somewhere in the main Methods texts that each intervention was repeated between 5-7 occasions in each monkey and not just in the legend please. This confused me for awhile and I wasn't really sure if they represented repeated injections or not. Alongside this, it is important that a table is provided that provides a list of the order of injections for each animal for each test so that the overall design of the entire study can be appreciated.

Whilst statistics has been performed on the modelled parameters of task performance for the DCZ and vehicle infusions in Fig 2, it is important that statistics is also performed on the actual data too please. Whilst individual t tests (without correcting for repeated tests) have been performed on each trial point a more appropriate analysis would be the use of an ANOVA with trial as an additional factor.

Why is the learning rate for the familiar reversal task so low? (Fig 2f) when the learning rates look comparable when comparing Fig 2c with Fig 2d, albeit the number of actual trials is considerably lower for the reversals? Surely, if anything, reversal learning rate should be greater for reversal in the familiar task in this case?

Why is the learning rate for the repeated sessions (individual dots) in Fig 2e and 2f for controls (blue bars) so variable?

Extended data Fig 4: Did the animals in 4b (pre-vector) get DCZ injections before the start of the session? To avoid confusion with 4a perhaps the DCZ injection needle (red arrow) could be placed at the start of the session? I originally thought this was the same expt as 4a except without hM4Di.

How many individual events are there for each of the surprising positive and negative experiences that are used for the analysis of the Novel task in Fig 4? How was the optimal rate calculated for this analysis? Optimal rate is explained in the earlier analyses but how was it calculated when only surprising positive and negative experiences are looked at separately?

Discussion:

On line 292/293 it is stated that 'OFC is needed for stimulus/action and outcomes' and reference one monkey study and one rodent study. But it is not clear that action-outcome associations are dependent upon OFC in monkeys. No action related encoding has been found in the OFC of monkeys.

Line 306. I don't think this is the correct reference? It doesn't look at WCST performance? The WCST specifically investigates attentional set-shifting?

Methods:

Were the monkeys experimentally naïve at the start of the study? If not, a description of what training/testing they had received prior to this should be provided as this could affect how the monkeys learned these two tests.

"Totals of 54 μ L and 50 μ L were injected for MK#1 and MK#2 via 53 and 49 tracks, respectively". Was this volume and number of injection sites per hemisphere or across both hemispheres?

Can the procedure for devaluation of the raisins/peanuts be described in full please? How long were they given free access to the food?

Can the description of the WCST be provided please in this manuscript.

Reviewer #3 (Remarks to the Author):

In this manuscript, the authors used PET-guided chemogenetic synaptic silencing to inhibit the orbitofrontal cortex (OFC) to rostromedial caudate (rmCD) pathway and the OFC to mediodorsal thalamus (MDm) pathway while nonhuman primates performed a task that relied on experience- and inference-based updating of stimulus-reward associations. They show that the OFC-rmCD mediates experience-based updating while the OFC-MDm pathway mediates inference-based updating. Furthermore, they suggest that the OFC mediates behavioral adaptation by regulating the switch between relying more heavily on experience- or inference-based learning. This is a nearly flawless demonstration of how differential pathways originating in the OFC contribute to value-based decision-making and learning from an elite group of nonhuman primate researchers pushing the boundaries of genetic tool use in nonhuman primates. The manuscript, however, would benefit from greater consideration of how chemogenetic expression aligns with known neuroanatomical details about OFC inputs to the striatum and amygdala in both primates and rodents to better situate their findings within the preexisting literature.

Strengths:

A major strength is that this paper uses a novel and complicated technique with chemogenetics to answer a simple question based on our current knowledge of OFC functionality in nonhuman primates. Furthermore, they use computational strategies to quantify inference- vs. experience-based learning in the behavior in the Familiar vs. Novel tasks.

I also appreciate the multiple controls they used for the behavior such as the WCST and the two-arm reversal learning task. In addition, the controls for the DCZ administration.

Weaknesses/Concerns:

The reported effects on the local infusion of DCZ are compelling, but it's surprising that inhibiting terminal projections would have a comparable effect to the inhibition of the cell bodies of those projections. Could the authors determine what proportion of variance relative to the effect of inhibiting the entire area is explained when either projection terminal is inhibited?

Another concern I have is why certain areas – like the amygdala – did not show any DREADD-positive areas even though it receives input from the OFC. Can the authors explain this better in terms of how the PET signals in OFC align with the known density of inputs to the amygdala? I think this is important because lesion work in macaques has shown that the amygdala and orbitofrontal cortex are necessary for inference-based reversal learning (Jang et al., 2015) but interestingly lesions of OFC areas 11/13 lead to inference-based impairments in reversal learning (particularly with OFC aspiration lesions which would damage projections targeted here), whereas lesions of the amygdala lead to inference-based enhancements due to reductions in the amount of evidence suggesting a reversal has occurred. Therefore, I think the authors should broaden their interpretation of their effects relative to other circuits known to be involved in inference-based learning and decision-making.

This leads to a more general comment that the manuscript would benefit from linking the PET signals at either terminals or cell bodies to known neuroanatomical details. For example, do the striatal projection zones for OFC cells expressing DREADDs seen here in caudate at all overlap with the known overlap in the projection zones of OFC and dorsal prefrontal regions (Averbeck et al., 2014)? Also, the reversal learning paradigms examined here have also been extensively studied in rodents.. Given the homologies between striatal and orbitofrontal circuits (Heilbronner et al., 2016), a broader discussion based on the neuroanatomical circuits is represents an opportunity to make contact with a broader literature examining similar circuits in rodents. This is warranted given that related circuit manipulations are known to affect reinforcement-based reversal learning (e.g., Groman et al. 2019).

It isn't clear to me why the data in which the putamen terminals are inhibited is not featured in the main manuscript but instead appears at the end of the supplemental data. The lack of an effect highlights the specificity of the potential circuitry involved.

The authors might want to address why they chose not to infuse DCZ into a non-DREADD-positive area that receives OFC input.

Reviewer: 1

In this article by Oyama et al., the authors applied their state-of-art imaging guided pathway-selective manipulation technique with DREADD and dissociated the functions of OFC-rmCD pathway and OFC-MDm pathway; that is, the former pathway regulates the experience-based value updating, while the latter pathway regulates inference-based value updating by using namely the NOVEL task and FAMILIAR task. The authors also fitted the behavioral data with two types of reinforcement learning models and empowered this hypothesis. This study is further extension of their 2021 paper on functional dissociation of dlPFC-CD and dlPFC-MD pathways, which also dissociated the functions of the two pathways originating from the same cortical regions. The experiments were carefully designed and results were clear and striking. I celebrate the authors for their achievement but have some conceptual comments on the manuscript.

Response 1: Thank you for your supportive and thoughtful remarks.

Major comment;

The authors claimed that the NOVEL task is related to the experience-based value updating and FAMILIAR task is related to the inference-based value updating. But to me, there seems to be more than that. In the case of NOVEL task, DCZ administration did not affect the acquisition phase but suppressed the reversal phase. If the subject is completely naïve to the stimulus sets and the rule, the reversal should also be relearning of the association of stimulus and reward from the beginning. If so, DCZ effect should be similar (no effect). But the present results indicate that the acquisition and reversal are different process, that is, the subject should have known the rule that the pattern (1,2,3,4,5) should be reversed to the pattern (5,4,3,2,1). This means there should be a process of strong inference in this NOVEL task, too. Thus, I doubt that the NAÏVE task is purely experience-dependent, free of inference. To me, it seems that the FAMILIAR task implements the low level (or more direct) inference, and the NOVEL task might implement the higher-order inference.

Response 2: Thank you for your comment. We appreciate the reviewer's emphasis on the differences between the acquisition and reversal phases in the NOVEL task. Indeed, our original interpretation may have been an oversimplification of the cognitive processes involved. The extensive training of the monkeys on the NOVEL task before the experimental sessions, as described in the manuscript, likely led to the development of a general understanding of the task structure, including the potential for reversal. The improved performance in the post-reversal phase, relative to the acquisition phase, suggests a cognitive ability that goes beyond purely new experience-dependent associations.

Therefore, as the reviewer suggests, it is worth considering that the monkeys might have used some degree of inference during the NOVEL task. As we show in the newly added “Supplementary Fig. 8”, the monkeys experienced the NOVEL task before the FAMILIAR task. This order implies that if some degree of inference was used in the NOVEL task, a similar cognitive strategy would also be used in the FAMILIAR task, given the monkeys' awareness of the possible reversal.

However, the effects of silencing the OFC-rmCD pathway in the NOVEL task, specifically during the reversal phase, did not match the effects observed for the FAMILIAR task. This leads us to a more straightforward interpretation: the critical involvement of the OFC-rmCD pathway in the NOVEL task is specific to the task context. Specifically, it appears to be essential for updating learned values based on experience following novel cue-reward associations. While this interpretation is straightforward, it does not rule out the possibility that some type of inference, such as awareness that a reversal could occur, could be a common element in both tasks.

To address the comments from the reviewer and to clarify our position, we have included a new paragraph in the Discussion section, detailing these considerations and acknowledging the potential for inference in both the NOVEL and FAMILIAR tasks. (lines 305-324)

Minor comments;

1. Figure 4 does not include the inset indicating what is red and what is blue.

Response 3: Thank you for pointing this out. We have added an inset indicating that red and blue represent DCZ and vehicle conditions, respectively.

2. In the legend of Extended Figure 1, there is no explanation of abbreviations PUT and AMG.

Response 4: We have now spelled out putamen and amygdala in the legend so that the abbreviations in the figure can be understood. (line 1099)

3. In the method (L646), the authors described the behavioral session began 30 min after the end of infusion. In the current experiments, the DCZ should have been diffused to the areas of larger volume (diameters of 5-7 mm or 3-4 mm). I wonder whether the effects were stable during the recording time. The authors should describe more about the recording time after infusion and how stable the effects were during the session.

Response 5: Thank you for your comment. After the infusion, each behavioral session (for both tasks) lasted approximately 1 hour. Thus, the session ended 90 min after the end of the infusion. This timeline has been explicitly detailed in the revised Methods section. (line 524-526)

To assess the stability of the effects observed after agonist infusion during the experimental sessions, we focused our analysis on the behavioral data in the FAMILIAR task, in which the task structure remains constant throughout the session (i.e., reversals of stimulus-reward contingencies were repeated). First, we divided the behavioral data into two phases: the first and second halves of the session. We then calculated the average optimal choice rates in five trials after the reversals, a period where behavioral deficits were observed in this task (please see also response #11 to reviewer #2). We conducted a three-way ANOVA (subject \times phase \times treatment) and found that while there was a significant main effect of treatment, the interaction between treatment and phase was not significant ($p = 0.79$). This suggests that the effects of local DCZ infusion remained consistent throughout the experimental sessions, spanning from 30 to 90 min post infusion.

To provide a clear visual representation of these assessments, we have included a new figure (Supplementary Fig. 6) and added a corresponding description in the Results section. (line 220-222)

Reviewer: 2

This study investigates the contribution of three pathways from OFC in experience-based and inference-based updating of stimulus-reward associations. Two monkeys were trained on two versions of a stimulus-reward learning and reversal task using either novel or familiar stimuli. DREADDs were then used to inactivate areas 11 and 13 as well as their separate pathways to the caudate, putamen and MD thalamus. OFC inactivation disrupted performance on both tasks whilst the pathway from OFC to MD disrupted the inference-based and not the experience-based task whilst the projection to the medial Caudate produced the opposite effect. Projections to the putamen had no effect on the novel task and so was not explored any further.

Response 1: We appreciate your interest and recognition of the importance of our findings.

The results are novel and of potential interest to the scientific community. However, the major drawback is that only two monkeys were used and the overall flavour of this study is that it was a pilot. Two is a very low 'n' for such intervention studies, even in non-human primates, and would usually be considered a pilot study, even more so since an important control manipulation, injecting the activator in the absence of DREADDs was only performed in one monkey, as was the use of the WCST to rule out effects of OFC on high cognitive load.⁽¹⁾ Moreover, the two monkeys received different viruses, one in which a single virus expressed both fluorescence and hM4Di and the other, in which the fluorescence was an independent virus.⁽²⁾ Given the concerns of replication failures across scientific studies it is unclear why only two monkeys were used in this intervention study? Whilst each intervention on each task was repeated on 5-7 occasions in each monkey this nevertheless does not help show the consistency of effects across monkeys, a very important consideration.⁽³⁾ Related to this is the fact that it is unclear whether these animals were experimentally naïve before the start of the study as past history can impact how an animal may learn behavioural tasks.⁽⁴⁾ Thus, overall, the results of this study need replicating in additional monkeys, preferably using the same viral vectors.⁽⁵⁾

Response 2: Thank you for your comment. The reviewer raised four major concerns, and thus we have numbered each one (in parentheses) and provide a corresponding response below.

(1) However, the major drawback is that only two monkeys were used... on high cognitive load.

We understand and acknowledge the reviewer's concerns about the sample size and its perception as a pilot study. However, we would like to offer an alternative perspective, which we believe the scientific community supports.

First, it is important to note that there is precedent in the field of behavioral studies using chemogenetics in monkeys for drawing robust conclusions from studies with small sample sizes (i.e., two monkeys), as evidenced by several publications (e.g., Raper et al., *eNeuro* 2019; Eldridge et al., *Nat Neuroscience* 2016; Oguchi et al., *Comm Biology* 2021; Noritake et al., *Nat Communications* 2023).

In our study, the strength of the conclusions about the functional roles of the OFC and its projections to two subcortical areas is supported by statistically significant and consistent findings across multiple experiments in these two monkeys. Our chemogenetic approach allows for a rigorous within-subject design, in contrast to ablation techniques that generally require larger sample sizes (typically three or more subjects) due to individual variability. This design allows us to study the effect of region/pathway manipulations in the same animals.

Another perspective is that, given the reliable reproducibility of the data from the two animals, and in light of established precedents, further animal testing would not be permitted due to ethical considerations to minimize animal use.

With regards DCZ control injections, although only one monkey underwent systemic pre-virus DCZ testing, both monkeys received several local DCZ injections as control for the pathway specific silencing, therefore we do not view the lack of systemic pre-virus DCZ testing in the second monkey as a relevant confound. Nevertheless, we acknowledge the reviewer's point regarding the single-subject control experiments as an area for future confirmation, and we have added a discussion of this as a limitation in the Discussion section. (line 403-411)

(2) Moreover, the two monkeys received different viruses, ... an independent virus..

We acknowledge the use of different viral vector constructs/systems for the expression of fluorescence and hM4Di in our two monkeys — one involved co-expression of fluorescence and hM4Di, while the other used separate vectors. Despite these differences, our PET imaging data clearly demonstrated that the functional protein (hM4Di) was effectively localized to the injection site, the OFC (Brodmann's area 11/13), and its

projection sites (rmCD and MDm) in both monkeys. This is now illustrated in the revised Fig. 2, which we have added at the suggestion of reviewer #2 to include the images that show hM4Di expression in both monkeys.

The consistent localization and expression of hM4Di in both monkeys strongly support the reliability and comparability of our behavioral observations. The effects for both systemic silencing, as well as pathway-selective silencing, can be confidently attributed to the chemogenetic action of hM4Di. Thus, despite the use of different viral vector systems, our findings emphasize the comparability of the results between subjects.

(3) Given the concerns of replication failures... a very important consideration.

We appreciate the reviewer's concern about the replicability and consistency of effects across subjects in our study in the context of replication failures in scientific research. Our decision to use only two monkeys was driven by ethical considerations to minimize the use of animals, as well as the practical tendency and methodological validity of this type of research, as discussed above (1).

In direct response to the reviewer's concern about the consistency of effects across the two monkeys, we have included additional analyses in the revised manuscript. Specifically, we performed three-way ANOVAs (subject \times phase \times treatment) and two-way ANOVAs (subject \times treatment) on the behavioral performance and model parameters. This allows us to examine the effect of each type of silencing (this is also in response to the comment about the Results; please see below).

Our results from two-way ANOVAs, which examined potential interactions between treatment and monkey, revealed no significant interactions in most of our tests. This finding provides strong evidence that silencing effects were consistent across monkeys, reinforcing the reliability and generalizability of our results.

However, we did observe a few exceptions, now shown in Fig 3e and Supplementary Fig. 9a, where a significant interaction was found, although there were tendencies for treatment-related effects. We have carefully addressed these exceptions in our manuscript (lines 1055-1058, 1219-1223) and provide a balanced discussion (line 407).

(4) Related to this is the fact... how an animal may learn behavioural tasks.

The animals used in this study were experimentally naïve at the start of the experiments (please see the response #18 below regarding the Methods).

(5) Thus, overall, the results of this study need replicating in additional monkeys, preferably using the same viral vectors.

As discussed above, we respectfully offer an alternative perspective to the reviewer's suggestion. We recognize the importance of replication in scientific research. However, we believe that our current study, with its rigorous methodological validity and consistent/reliable DREADD expression (despite using different viral vectors, see response #2.2) and consistent results across multiple experiments in two monkeys (supported by additional analyses see response #2.3], already provides robust evidence to support our conclusions.

With regards the conclusions from this study, it would have been nice if there had been some discussion regarding the results in the context of the different neural pathways that have been implicated. Why for example would experience-based updating be dependent upon the caudate pathway whilst inference-based updating be more dependent on the MD thalamus pathway? What else is known about the role of the striatum and thalamus in such cognitive functions that may explain this differentiation?

Response 3: Thank you for your comment. As suggested, we have added a discussion about functional roles of the caudate and MD thalamus in experience- and inference-based updating in the Discussion section. (lines 332-346)

Moreover, it is stated in the final sentence that these findings are relevant to our understanding of neuropsychiatric disorders but why? Is there evidence of such specific, dissociable deficits in patients?

Response 4: Thank you for this important question. The relevance of our findings to neuropsychiatric disorders, particularly obsessive-compulsive disorder (OCD), is indeed a critical aspect of our discussion. In OCD, there is substantial evidence for abnormal functional connectivity between the OFC and the caudate nucleus. This is consistent with our findings and highlights the importance of these neural connections. To clarify this link, we have revised the Discussion section as follows:

“..., our findings have potential implications for understanding certain psychiatric disorders, such as obsessive-compulsive disorder (OCD). Our findings are particularly relevant to OCD because abnormal functional connectivity between the OFC and the caudate nucleus is frequently reported in patients with OCD⁶⁴”. (line 426-429)

It is also stated in the discussion that: “Although the OFC and dlPFC are located in different parallel cortico-basal ganglia-thalamo-cortical circuits, the observed similarity that different subcortical pathways have different functions suggests that the parallel prefrontal-subcortical networks might share a common neural basis for adaptive behavior”. Just because multiple pathways from distinct prefrontal regions e.g. dlPFC and OFC have been shown to perform different functions it is premature to suggest that they share a common neural basis for adaptive behaviour? In many ways it might be surprising if different pathways from the same region did perform the same function? Different pathways from many other cortical and subcortical regions have been shown to perform different functions so I don't think the finding says anything special about dlPFC and OFC and the parallel cortico-striatal pathways they contribute to?

Response 5: Thank you for your insightful feedback. We recognize that our original statement might have been overly broad in suggesting a common neural basis for adaptive behavior across various prefronto-subcortical pathways. We meant that our results, including those from previous studies, lead to the prediction that other prefronto-subcortical pathways, such as those from the ventrolateral PFC to the caudate and MD thalamus, might have distinct roles in cognitive functions related to each prefrontal origin. In light of your comment, we have rephrased our discussion to more accurately reflect this understanding: (lines 389-393)

“Given the parallel forms of cortico-basal ganglia-thalamo-cortical circuits⁶¹, these consistent observations lead us to predict that other prefronto-striatal and prefronto-thalamus pathways, e.g., pathways from the ventrolateral PFC to the caudate and MD thalamus, may also have specialized roles aligned with the cognitive functions associated with their prefrontal origin, as recently investigated in the rodents.”

Results.

Describing this region as lateral OFC is confusing as very often the neighbouring area 12/47 is very often called lateral OFC too. So, I recommend just referring to this area as OFC11-13 so as to avoid confusion.

Response 6: Thank you for pointing out the potential confusion in our terminology. We agree that it is confusing that we had referred to the area (Brodmann's area 11 and 13) as "lateral" OFC. In the revised manuscript, we have removed the term "lateral" when referring to this area. The manuscript now clearly states:

"First, we bilaterally introduced the inhibitory DREADD hM4Di into the OFC (Brodmann's area 11/13, hereafter referred to as "OFC11/13") of each monkey ..." (line 140-142).

In addition, we have made a similar adjustment in the context of discussing this area:

"... at least the area that we focused on in this study (Brodmann's area 11/13) ..." (line 366).

For consistency, and to avoid any further confusion with the hyphen often used to indicate connections between two areas or reference numbers, we have replaced "OFC" with "OFC^{11/13}" in the Results section and figures/legends. However, for simplicity and readability, we have retained the use of "OFC" in more general references, such as in the Abstract, Introduction, and Discussion sections.

In the figures it would be preferable to see the data for each monkey separately rather than averaging the two monkeys performance on repeated sessions. In Fig 1 the two monkeys could be depicted on the same graphs in 1b and c.

Response 7: In response to this suggestion, we have revised Figures 1, 2, and 3 to show the data for each monkey separately, and revised the corresponding legends. (line 993-995, 999, 1009, 1044)

Not clear what 'latter' is referring to on line 121. Page 6.

Response 8: Thank you for pointing this out. We have revised the sentence as follows:

"... for details, see the last paragraph of the Results section". (line 121)

For completeness/thoroughness, in Fig 2, can the bilateral PET signal in OFC be shown, rather than just a unilateral image please? Since only two monkeys took part in this study then it is important to show both monkey's data.

Response 9: In response to this suggestion, we have revised Fig. 2 to include the bilateral PET data for both monkeys separately, and revised the corresponding legends. (line 1005-1007)

Can an explicit statement be made somewhere in the main Methods texts that each intervention was repeated between 5-7 occasions in each monkey and not just in the legend please. This confused me for a while and I wasn't really sure if they represented repeated injections or not. Alongside this, it is important that a table is provided that provides a list of the order of injections for each animal for each test so that the overall design of the entire study can be appreciated.

Response 10: Thank you for this valuable suggestion. We have revised the figure legends and the Methods section to include an explicit statement about the frequency of interventions. (line 524-525, 1044)

We have also provided a new figure to show an overview of the experimental design (Supplementary Fig. 8), and revised the description about the order of the experiments in the Results and Methods sections. (line 230-231, 525-527)

Whilst statistics has been performed on the modelled parameters of task performance for the DCZ and vehicle infusions in Fig 2, it is important that statistics is also performed on the actual data too please. Whilst individual t tests (without correcting for repeated tests) have been performed on each trial point a more appropriate analysis would be the use of an ANOVA with trial as an additional factor.

Response 11: In response to your suggestion, we have conducted an ANOVA to more comprehensively analyze the actual task performance. Specifically, we divided each session into three phases: pre-reversal (Pre), the early part of the post-reversal (PoE), and the late part of the post-reversal (PoL). These phases consisted of 90 pre-reversal, the first 100 post-reversal trials,

and the next 110 trials in the NOVEL task, and 10 pre-reversal, the first 5 post-reversal and the next 5 post-reversal trials in the FAMILIAR task.

We then conducted a three-way ANOVA (subject \times phase \times treatment) to test for a significant difference in treatment (Vehicle vs. DCZ) and the interaction between phase and treatment. The purpose of this step was to determine whether deficits were more pronounced in a specific phase. Subsequently we conducted two-way ANOVAs (subject \times treatment) for each phase to determine in which phase(s) a significant difference in treatment occurred.

These analyses revealed that the deficits were restricted to the early phase after the reversal in both tasks. These results, along with a detailed analysis, are now presented in Fig. 2e, g and Fig. 3e, f. We have also included a comprehensive description of this analysis, mainly in each figure legend (lines 1014-1026, 1046-1065), and further detailed in the Methods section (lines 601-614).

Why is the learning rate for the familiar reversal task so low? (Fig 2f) when the learning rates look comparable when comparing Fig 2c with Fig 2d, albeit the number of actual trials is considerably lower for the reversals? Surely, if anything, reversal learning rate should be greater for reversal in the familiar task in this case?

Response 12: Thank you for asking this insightful question. In our study, we used different models to analyze the behavioral data of the two tasks. Specifically, we used the EXP model for the NOVEL task and the INF models for the FAMILIAR task. These models were chosen based on their suitability to the behavioral data, likely reflecting the specific characteristics of each task and the nature of the learning processes involved. Consequently, the learning rates derived from these two tasks should not be directly compared due to the different mathematical frameworks and assumptions underlying each model. To address the potential confusion this might cause, we have added a clarifying note to the legend of Fig. 2 as follows:

“Note that the learning rates for the NOVEL and FAMILIAR tasks were calculated using different models. Thus, they are not directly comparable.” (line 1031-1033)

Why is the learning rate for the repeated sessions (individual dots) in Fig 2e and 2f for controls (blue bars) so variable?

Response 13: Thank you for your comment. As the reviewer pointed out, the distribution of learning rates looks variable, especially in the vehicle condition. However, such variability in distribution is not uncommon in studies involving animal behavior, especially in those employing reinforcement learning models. This has been documented in previous research; for example, Fig. 3e in Wittmann et al., *Nature Communications*, 11, 3771 (2020) and Fig. 4C in Huang et al., *Journal of Neuroscience*, 43(10):1714–1730 (2023). Both report similar findings with non-Gaussian learning rate distributions.

When we looked more closely at our data, particularly at the sessions with higher learning rates (1 session of Mk#2 in the NOVEL task, and 1 session of Mk#1 and 2 sessions of Mk#2 in the FAMILIAR task, which corresponding to dots with higher values in Fig. 2f, h), a trend became apparent. In these sessions, the monkeys generally showed higher performance, including higher optimal choice rates immediately after the reversal in the NOVEL task, and small numbers of trials needed to reach performance plateaus in the FAMILIAR task. Such improved performances likely contributed to the higher learning rates.

Extended data Fig 4: Did the animals in 4b (pre-vector) get DCZ injections before the start of the session? To avoid confusion with 4a perhaps the DCZ injection needle (red arrow) could be placed at the start of the session? I originally thought this was the same expt as 4a except without hM4Di.

Response 14: Thank you for your comment. To address your concern, we have made several revisions to ensure clarity in how we described the timing of DCZ injections in the experiments, now shown in Supplementary Fig. 3. In this figure, we have added a symbol representing an injection needle along with a red arrow to indicate the timing of the DCZ injection, which was 15 min before the beginning of the experiments. Correspondingly, the figure legend has been revised as follows:

“DCZ was administered 15 min before the beginning of the experiments as with other systemic injections”. (line 1137)

To further clarify the overall procedures, we have revised the description in the Methods section about the timing of the systemic injection:

“For systemic intramuscular injection, this stock solution was first diluted in saline to a final concentration of 1 mg/mL, and a dose of 100 µg/kg was injected 15 min before the beginning of the experiments, unless otherwise noted.” (line 497-500)

In addition, we have made this figure the same style as other similar figures.

How many individual events are there for each of the surprising positive and negative experiences that are used for the analysis of the Novel task in Fig 4? How was the optimal rate calculated for this analysis? Optimal rate is explained in the earlier analyses but how was it calculated when only surprising positive and negative experiences are looked at separately?

Response 15: Thank you for your insightful query. To clarify, in our analysis for Fig. 4, which focused on the Novel task, we observed an average of 7.5 instances (ranging from 5 to 11) for unpredicted positive experiences and approximately 6.5 instances (ranging from 5 to 9) for unpredicted negative experiences during the systemic injections.

For a learning-model agnostic analysis (Fig. 4), we defined “unpredicted positive experience” as a scenario in which a choice was made for a higher valued options (4 or 5 drops) following a trial in which this option was not chosen despite being available within the PoE time window (see the comment above). Conversely, an “unpredicted negative experience” was identified when a lower valued option (1 or 2 drops) was selected following a trial where this lower option was not chosen.

To quantitatively assess the impact of these experiences on decision making, we calculated the averaged optimal choice rate in subsequent trials that presented the higher valued options or lower valued options for each scenario, respectively.

For example: The monkey is presented with “5-drops-of-juice” and “2-drops-of-juice” options but does not choose the 5-drops option. Subsequently, it is presented with the choice between 3-drops and 5-drops stimuli and chooses the 5-drops option (this is the unpredicted positive experience). A future trial in which the monkey has to again choose between the 2-drops and 5-drops options would be the one for which the optimal choice rate was calculated.

Throughout the manuscript, the calculation of the optimal choice rate was always the rate of choosing the option associated with higher rewards, regardless of whether the outcome of the option was surprising to monkeys, i.e., an unexperienced stimulus-reward association. To increase

the clarity and readability of our methods, we have included this description in the Methods section. (line 620-636)

Discussion:

On line 292/293 it is stated that ‘OFC is needed for stimulus/action and outcomes’ and reference one monkey study and one rodent study. But it is not clear that action-outcome associations are dependent upon OFC in monkeys. No action related encoding has been found in the OFC of monkeys.

Response 16: Thank you for pointing this out. We have rephrased this sentence as follows:

“... associations between stimuli and outcomes”. (line 349)

Line 306. I don’t think this is the correct reference? It doesn’t look at WCST performance? The WCST specifically investigates attentional set-shifting?

Response 17: We have updated the reference to accurately reflect relevant WCST research: the correct citation now refers to Buckley et al. (2009), which reported lesion experiments in a monkey performing a modified WCST. (line 363)

Methods:

Were the monkeys experimentally naïve at the start of the study? If not, a description of what training/testing they had received prior to this should be provided as this could affect how the monkeys learned these two tests.

Response 18: Yes. The two monkeys used were experimentally naïve at the start of the study. We have revised the Methods to include this information as follows:

“Two experimentally naïve male Japanese monkeys ...”. (line 432)

*“Totals of 54 μ L and 50 μ L were injected for MK#1 and MK#2 via 53 and 49 tracks, respectively”.
Was this volume and number of injection sites per hemisphere or across both hemispheres?*

Response 19: Thank you for your attention to this detail. We injected totals of 54 μ L and 50 μ L (1 μ L/track) for the left and right hemispheres of MK#1, respectively, and 53 μ L and 49 μ L for the left and right hemispheres of MK#2, respectively. We have revised the description in Methods as follows:

“For Mk#1, total volumes of 54 μ L and 50 μ L were injected via 54 and 50 tracks in the left and right hemispheres, respectively. For Mk#2, total volumes of 53 μ L and 49 μ L were injected via 50 and 47 tracks in the left and right hemispheres, respectively.” (line 473-476)

Can the procedure for devaluation of the raisins/peanuts be described in full please? How long were they given free access to the food?

Response 20: We allowed the monkey to freely access either food for 30 min as previously described (e.g., Rudebeck et al., 2013). We have revised the description in Methods as follows:

“For devaluation, the monkey was given up to 30 min to consume as much of either food as it wanted. The devaluation procedure was deemed to be complete when the monkey refrained from retrieving food from the food chamber for 5 min.” (line 570-573)

Can the description of the WCST be provided please in this manuscript.

Response 20: Thank you for your comment. We have added a description of the monkey WCST in the Methods. (line 580-598)

Reviewer: 3

In this manuscript, the authors used PET-guided chemogenetic synaptic silencing to inhibit the orbitofrontal cortex (OFC) to rostromedial caudate (rmCD) pathway and the OFC to mediodorsal thalamus (MDm) pathway while nonhuman primates performed a task that relied on experience- and inference-based updating of stimulus-reward associations. They show that the OFC-rmCD mediates experience-based updating while the OFC-MDm pathway mediates inference-based updating. Furthermore, they suggest that the OFC mediates behavioral adaptation by regulating the switch between relying more heavily on experience- or inference-based learning. This is a nearly flawless demonstration of how differential pathways originating in the OFC contribute to value-based decision-making and learning from an elite group of nonhuman primate researchers pushing the boundaries of genetic tool use in nonhuman primates. The manuscript, however, would benefit from greater consideration of how chemogenetic expression aligns with known neuroanatomical details about OFC inputs to the striatum and amygdala in both primates and rodents to better situate their findings within the preexisting literature.

Response 1: We are grateful for your appreciative remarks.

Strengths:

A major strength is that this paper uses a novel and complicated technique with chemogenetics to answer a simple question based on our current knowledge of OFC functionality in nonhuman primates. Furthermore, they use computational strategies to quantify inference- vs. experience-based learning in the behavior in the Familiar vs. Novel tasks.

I also appreciate the multiple controls they used for the behavior such as the WCST and the two-arm reversal learning task. In addition, the controls for the DCZ administration.

Response 2: Thank you for your positive feedback.

Weaknesses/Concerns:

The reported effects on the local infusion of DCZ are compelling, but it's surprising that inhibiting terminal projections would have a comparable effect to the inhibition of the cell bodies of those projections. Could the authors determine what proportion of variance relative to the effect of inhibiting the entire area is explained when either projection terminal is inhibited?

Response 3: Thank you for your comment. While a direct comparison between systemic injection and local infusion experiments is challenging because they were conducted in different times (please see Supplementary Fig. 7), we have attempted to quantify the effect size of each approach. We calculated the effect size for both systemic injection and local infusions within the PoE time-window (which includes 100 post-reversal trials in the NOVEL task and 5 post-reversal trials in the FAMILIAR task; see also the response #11 to reviewer #2). This was achieved through a two-way ANOVA (subject \times treatment). Analysis showed that the effect size of systemic injections and local infusions were almost comparable in the NOVEL task (0.50 vs. 0.48 for systemic injections and local infusions into the rmCD). However, in the FAMILIAR task, the effect size was notably larger for systemic injections (0.46) compared with that for local infusions into the MDm (0.29). This suggests that the OFC-rmCD pathway is critically involved in implementing experience-based updating. Nevertheless, we do not think that this excludes the possibility that other regions/pathways are critically involved. It should also be noted that our methods, which rely on the diffusion of agonist solution and the proportion of neurons expressing DREADDs, do not allow us to definitively determine whether we have inhibited an entire subset of projections from the OFC.

In response to the comments, we have added details about these effect size results in the Results section (line 215, 219), and further elaborated on this topic in the Discussion section. (line 411-418)

Another concern I have is why certain areas – like the amygdala – did not show any DREADD-positive areas even though it receives input from the OFC. Can the authors explain this better in terms of how the PET signals in OFC align with the known density of inputs to the amygdala? I think this is important because lesion work in macaques has shown that the amygdala and orbitofrontal cortex are necessary for inference-based reversal learning (Jang et al., 2015) but interestingly lesions of OFC areas 11/13 lead to inference-based impairments in reversal learning (particularly with OFC aspiration lesions which would damage projections targeted here), whereas lesions of the amygdala lead to inference-based enhancements due to reductions in the amount of evidence suggesting a reversal has occurred. Therefore, I think the authors should broaden their interpretation of their effects relative to other circuits known to be involved in inference-based learning and decision-making.

Response 4: Thank you for raising this insightful concern. Previous anatomical studies have shown that the projections from the OFC to the amygdala arise predominantly from the caudal

part of the orbital area. These studies highlight stronger projections from the caudal part and relatively weaker projections from anterior orbitofrontal area (Ghashghaei & Barbas, *Neuroscience* 115, 4 (2002); Ghashghaei et al., *Neuroimage*, 34 (2007)). In our experiments, we injected vectors into the relatively anterior part (Brodmann's area 11/13), which could explain the lack of significant PET signal in the amygdala. In addition, our current protocol may not be sensitive enough to detect sparser projections, such as cortico-cortical projections.

However, we acknowledge that this does not exclude the possible involvement of the amygdala in experience- and/or inference-based updating, as evidenced by previous studies. It is conceivable that future studies, especially those that target the silencing the caudal part of the OFC and its associated circuits, like the OFC-amygdala pathway, would provide insight into the similarity and differences in functional roles in value-based behavior.

In response to the comments, we have expanded a discussion about the amygdala in relation to our study in the Discussion section. (line 370-384)

This leads to a more general comment that the manuscript would benefit from linking the PET signals at either terminals or cell bodies to known neuroanatomical details. For example, do the striatal projection zones for OFC cells expressing DREADDs seen here in caudate at all overlap with the known overlap in the projection zones of OFC and dorsal prefrontal regions (Averbeck et al., 2014)?

Response 5: Thank you for your thoughtful comment emphasizing the importance of correlating our PET signal with established neuroanatomical details. As we have shown previously (Nagai et al., 2020, *Nature Neuroscience*; Oyama et al., 2021, *Science Advances*) and also in this study, the PET signal nicely corresponds to the actual histological data. To directly address the question raised by the reviewer, we combined the two datasets (OFC^{11/13}, from this study and the dIPFC, from Oyama et al., 2021) to investigate potential overlap in the projection zones. By fitting the PET data from each animal to the standard brain, we were able to observe these projection patterns (see the figure below; please note that the scale of the PET signal has been adjusted to highlight areas of overlap). This indicates a notable overlap in the ventromedial caudate, which is consistent with the results reported by Averbeck et al. (2014). We have added a description of this overlap of projections from the OFC and dIPFC, and provided a link between the functional implication of the "convergence zone" and our findings in the Discussion section. (line 394-399)

While we continue to develop/refine our protocol for the quantitative evaluation of PET signals, these preliminary results will be published elsewhere.

PET signal around the caudate nucleus for hM4Di fitted and superimposed on the standard brain atlas. Green, vector injections into the dlPFC (data from Oyama et al., 2021); blue, injections into the OFC in Mk#2 of this study. The overlapping areas are indicated by red dashed lines.

Also, the reversal learning paradigms examined here have also been extensively studied in rodents.. Given the homologies between striatal and orbitofrontal circuits (Heilbronner et al., 2016), a broader discussion based on the neuroanatomical circuits is represents an opportunity to make contact with a broader literature examining similar circuits in rodents. This is warranted given that related circuit manipulations are known to affect reinforcement-based reversal learning (e.g., Groman et al. 2019).

Response 6: Thank you for your suggestion. We agree that integrating a broader literature, including important rodent studies, can enrich our discussion on the functional role of prefronto-subcortical circuits. We have now included additional references for various contexts in the overall Discussion section (e.g., reversal learning and pathway-selective manipulation) (e.g., line 336-346)

It isn't clear to me why the data in which the putamen terminals are inhibited is not featured in the main manuscript but instead appears at the end of the supplemental data. The lack of an effect highlights the specificity of the potential circuitry involved.

The authors might want to address why they chose not to infuse DCZ into a non-DREADD-positive area that receives OFC input.

Response 7: Thank you for your constructive suggestion. There were two key considerations in the decision to place these data in the Supplementary section rather than the main manuscript. First, the experiments for silencing the OFC-putamen pathway were not performed in the FAMILIAR task. Second, the data were obtained only from Mk#2. To make this clearer, we have now added a specific mention in the manuscript that the results are derived from Mk#2 only as follows:

“...and behavioral performance (right) obtained from Mk#2”. (line 1179)

In addition, as the reviewer pointed out, induction of DCZ into a non-DREADD-area would be an important control for the specificity of our findings. Indeed, we have conducted this control previously (Oyama et al., 2021, Science Advances) and in the current study in one monkey (Mk#1). In this control, during the performance of the NOVEL task, we infused DCZ into the anterior thalamic nucleus that is located between the rmCD and MDm. We found no significant deficits. Although we did not present these data in the original manuscript for simplicity, we have now included them in the revised manuscript as Supplementary Fig. 7c-d and added a corresponding description in the Results section. (line 225-229)

REVIEWERS' COMMENTS

Reviewer #1 (Remarks to the Author):

In the revised manuscript, I confirmed that the authors appropriately responded to my comments on the previous version. Therefore, I have no more comments on this manuscript.

Reviewer #2 (Remarks to the Author):

The Reviewers have comprehensively addressed all my queries including the low numbers of monkeys used. Having clarified that each monkey had 5-6 repeat infusions I am satisfied with the robustness of their findings. I also think it is important that they have openly discussed this issue under limitations in the discussion.

My one final query was whether the authors could directly compare the pathway specific effects of DCZ on the two tests? They've shown that one OFC pathway differentially impacts one task and not the other in one analysis and vice versa for the other pathway. Can they demonstrate in a single analysis that the effect of the two pathways is statistically different from one another? I don't think this comparison has been made but correct me if I'm wrong?

Reviewer #3 (Remarks to the Author):

The authors have adequately addressed all of my previous concerns. I especially thank them for the preliminary data aligning the PET signals with histology and known neuroanatomical details. This remains an exciting contribution.

Reviewer #2 (Remarks to the Author):

The Reviewers have comprehensively addressed all my queries including the low numbers of monkeys used. Having clarified that each monkey had 5-6 repeat infusions I am satisfied with the robustness of their findings. I also think it is important that they have openly discussed this issue under limitations in the discussion.

My one final query was whether the authors could directly compare the pathway specific effects of DCZ on the two tests? They've shown that one OFC pathway differentially impacts one task and not the other in one analysis and vice versa for the other pathway. Can they demonstrate in a single analysis that the effect of the two pathways is statistically different from one another? I don't think this comparison has been made but correct me if I'm wrong?

R: We thank the reviewer's constructive feedback and the final query regarding a direct comparison of the silencing effects between two different pathways in a single analysis. In response to this, we have performed an additional analysis using three-way ANOVA (factors: monkey x treatment x injection area) during the Po1 phase. This revealed significant interactions between treatment and injection area for both tasks (NOVEL, $F(1,32) = 6.1$, $p = 0.019$; FAMILIAR, $F(1,32) = 6.6$, $p = 0.015$). These results suggest that the effects of DCZ infusion into different areas are statistically different, supporting the double dissociation of pathway-specific manipulations. We have included these findings into the legend of Figure 3.